

# Inferring the tree regeneration niche from inventory data using a dynamic forest model

Yannek Käber[1,2], Florian Hartig[3], Harald Bugmann[1]

[1]Forest Ecology, Institute of Terrestrial Ecosystems, Department of Environmental Systems Science, ETH Zurich, Zurich, 8092, Switzerland
[2]Chair of Silviculture, Faculty of Environment and Natural Resources, University of Freiburg, Freiburg, 79106, Germany
[3]Theoretical Ecology Lab, Faculty of Biology and Pre-Clinical Medicine, University of Regensburg, Regensburg, 93053, Germany

*Correspondence to*: Yannek Käber (y.kaeber@posteo.de)

**Abstract.** The regeneration niche of trees is governed by many processes and factors that are challenging to determine. Besides species distribution, which determine if seeds are available, complex local dynamics in forest ecosystems (e.g., competition, pathogens) exert fundamental influence on tree regeneration. Consequently, the representation of tree regeneration in dynamic forest models is a notoriously complicated process which often involves many subprocesses. The ForClim forest gap model described regeneration mainly by species traits and the ecological knowledge linking these traits to regeneration properties. However, this regeneration module was never validated with large-scale data. Here, we compare this trait-based approach with an alternative, namely an inverse calibration approach, where we estimate regeneration parameters from a large observational dataset of unmanaged European forests. The model inversion was done using Bayesian inference with a simple and complex model variant without and with competition during regeneration. In this approach, we estimate shade and drought tolerance as well as the temperature requirements for 11 common tree species along with the intensity of regeneration (i.e., the maximum regeneration rate). We find that the parameters determining species' light niche (i.e., light requirements) are similar between the trait based and calibrated values for both model variants, but only the complex model led to plausible estimates of the drought niche. The temperature niche as defined in by traits could not be recovered from the data by either model variant using inverse calibration. The parameter estimates differed between the complex and the simple model, with the complex model performing better. In both model variants, the calibration strongly changed the parameters that determine regeneration intensity compared to the default.

We conclude that the regeneration niche of the tree species in this large European dataset can be recovered in terms of the stand-level parameters light availability and regeneration intensity, while abiotic drivers (temperature and drought) are more elusive. The higher performance of the inversely calibrated models underpins the importance of informing dynamic models by real-world observations. Future research should focus on an even larger environmental coverage of observations of demographic processes in unmanaged forests to verify our findings at species range limits under extreme climatic conditions.



## Introduction

Predictions of species range shifts and forest dynamics under climate change require process-based models that account for
the complex feedback between stand dynamics, soils, and climate (Morin and Thuiller, 2009). In this context, tree regeneration
is a particularly important (McDowell et al., 2020; Hanbury-Brown et al., 2022), but also quite uncertain process (Price et al.,
2001; König et al., 2022), as shown by numerous studies that yield different results depending on the site, species, and spatial
and temporal scale that is considered (Clark et al., 1999; Lett and Dorrepaal, 2018). The reason for these divergences among
observations and experimentation is a) that many seemingly stochastic regeneration processes are actually controlled by biotic
and abiotic conditions that vary across a wide range of temporal and spatial scales (Grubb, 1977; Hart et al., 2017), coupled
with b) the lack of suitable data to consistently study these processes on different scales (Clark et al., 1999).

A challenge that hinders progress on these questions is that a tree's regeneration niche may be distinct from the habitat niche
(physical limits of adults), the life-form niche (size and productivity), and the phenological niche (seasonal development)
(Grubb, 1977). The regeneration niche is defined as the set of factors that increase the chance for successful replacement of
one adult by a new adult of the next generation. Together, these niche types characterize a plant's full niche (Grubb, 1977).
The distinction between the regeneration niche vs. niche types relevant to adults is helpful to understand that while certain
environmental drivers are fundamental for a plant's niche, their importance potentially differs significantly between life stages
(cf. the concept of ontogenetic niche shifts; Werner & Gilliam, 1984).

Differences between the niches of young trees and adults are evident e.g. from divergent occurrences along large-scale climatic
gradients (e.g., Lenoir et al., 2009; Mathys et al., 2018; Wason & Dovciak, 2017; Woodall et al., 2009), but it is not clear
whether these differences are caused by climate change (Smithers et al., 2018) or ontogenetic niche shifts (cf. Heiland et al.,
2022). Together with climatic drivers, small-scaled processes such as competition for light (Collins and Good, 1987; Poorter,
2007) or biotic interactions (HilleRisLambers et al., 2013; Käber et al., 2023a) are key determinants of a species regeneration
niche.

To assess the different niche processes, a possible way forward is to disentangle the effects of large- and small-scaled
environmental drivers on tree regeneration in a dynamic forest model (DFM) based on regeneration data that stem from natural
conditions such as unmanaged forests. Combining DFMs and data from unmanaged forests can be used to tackle yet unresolved
issues of analysing and modelling tree regeneration. First, DFMs represent complex stand-environment feedbacks explicitly,
which puts the quantified effects in the context of specific processes. Thus, it opens up opportunities for more nuanced
inference on processes instead of yielding indistinct associations between observed regeneration patterns and environmental
drivers. Second, using data from unmanaged forests minimizes the confounding influence of management on the natural
dynamics of demographic processes. Specifically, promotion of certain species or individual trees through planting or thinning
are absent in unmanaged forests.

Regeneration in most DFMs is captured relatively simply (compared to growth and mortality) and, due to the lack of detailed
process data, mostly phenomenologically (König et al., 2022). Nevertheless, even these simplified representations of tree



regeneration are characterized by widely different levels of complexity (Bugmann and Seidl, 2022). Typically, the representations of regeneration in DFMs are based on knowledge that is abstracted to an aggregate over many processes (Price et al., 2001). This leads to high stochasticity and renders the validation of tree regeneration patterns in DFMs challenging.

Over the past decade, robust methods for evaluating stochastic models of ecological processes with data have been developed
(cf. Hartig et al., 2011). Yet, only a few studies have confronted tree regeneration models with forest inventory data (Diaz-Yanez et al. in review; Rüger et al., 2009). An important reason is the issue of elucidating the drivers of ecological processes at different spatial and temporal scales mechanistically, as the apparent stochasticity makes it challenging to retrieve signal from the data (Hart et al., 2017; Oberpriller et al., 2021; Shoemaker et al., 2020). Specifically, trade-offs between meaningful observations for key small-scale processes and the coverage of wide abiotic gradients impede a comprehensive analysis across
the stages of tree regeneration (Clark et al., 1999). Consequently, there is a need to advance the frontier of evaluating tree regeneration in DFMs with data (cf. Diaz-Yanez et al. in review).

In the DFM ForClim (Huber et al., 2020), which we use as a case study, the regeneration niche is defined, among others, based on light availability, water availability, and summer temperature conditions. The niche is defined by trait values for shade tolerance, drought tolerance, and minimum degree-days (e.g., Leuschner and Ellenberg, 2017). The model is highly sensitive
to the values of these parameters (cf. Huber et al., 2018), which adds weight to their detailed evaluation based on real-world observations. Indeed, there are multiple expectations regarding the usefulness of trait values in DFMs: generally, species traits are most suitable to model forest dynamics if their ecological context is clearly defined, i.e., traits that are tied to a specific process which is explicitly modeled are expected to have more predictive power than traits that act on multiple processes which are not considered individually (Yang et al., 2018). Functional traits have successfully been applied in a simple dynamic model
of annual plant communities (Chalmandrier et al., 2021), thus underpinning the validity of trait-based approaches for modeling plant demography. In ForClim, the interplay between the traits of multiple species is implemented in a simple variant of the regeneration model that does not consider competition and a more complex model variant that includes it (Huber et al., 2020). Interestingly, in both variants the link between traits and processes leads to ecologically plausible emergent properties of simulated potential natural vegetation along elevational gradients in the Swiss Alps (Huber et al., 2020) and elsewhere
(Bugmann and Solomon, 2000), yet it remains unclear whether the simulated regeneration processes (i.e., ingrowth rates) are actually in agreement with observational data (Zell et al., 2019; Käber et al., 2021).

Here, we evaluate possible reasons for mismatches between process formulations and observations by comparing two approaches for parameterizing the regeneration niche in ForClim: (1) a trait-based approach, where the regeneration niche is based on trait values determined *a priori* from ecological knowledge, and (2) an inverse calibration approach, where the trait
values are derived *a posteriori* using a novel observational data set of demographic processes in European unmanaged forests that covers unprecedented spatial and temporal scales (Käber et al., 2023a). Specifically, we address two research questions.

- How does the regeneration niche that is emerging from the inverse calibration differ compared to the niche defined by species' traits?



- Does a more complex regeneration model that includes competition feature a higher performance compared to a
simple regeneration model without competition?

**Methods**

**Data**

**Forest inventory data**

We used records of tree recruitment from 6,540 forest inventory plots covering 299 strict forest reserves that are curated by 18
European research institutions in the context of the European Forest Research Initiative (EuFoRIa, www.euforia-project.org)
(Käber et al., 2023a). Depending on the forest inventory design, different diameter thresholds (DBH, diameter at breast height)
were used as the callipering limit in the inventories (i.e., 4, 7 or 10 cm). These inventory plots were aggregated or split into
units of ca. 1 ha to obtain samples of similar spatial extent (Käber et al., 2023a) that reduce the stochasticity in the data and
thus increase the stability of the signal used for model evaluation. After data processing (cf., Käber et al., 2023), 865 plots
were available for this study. Some trees within these plots had implausible DBH measurements (e.g., growth was
unrealistically high or negative), which required the exclusion of some plots.

We defined two criteria for selecting plots suitable for the study. The first criterion evaluated the number of trees with
implausible measurements relative to the total number of measured trees and observed basal area: at least 95 % of all trees
needed to have plausible measurements together with at least 95 % of the basal area being comprised of trees with plausible
DBH measurements. The second criterion evaluated the number of trees with implausible measurements relative to the plot
area: the maximum number of trees with implausible measurements per ha allowed in the data set was defined as the 75th
percentile of trees with implausible measurements per ha (which amounted to 14.31 ha$^{-1}$). The second criterion was defined
because some plots did not fulfil the first criterion although they had a relatively low number of trees with implausible
measurements per ha. These were particularly large plots with low tree number per ha (N = 51). After plot selection, all trees
with implausible DBH measurements of the selected plots were removed.

The final number of sites considered was 696 with sufficient information on tree regeneration for 11 tree species. All other
species were aggregated in an extra category ("other" species). About half of the sites (353) were used for calibration (training
data), and the other half (343) for evaluation (test data). We split the sites so that the variation of the represented inventory
data sets (i.e., the individual data associated to one research institution) and DBH thresholds was similar in both test and
training data. This also resulted in similar variations of environmental conditions because each inventory data set (from a given
institution) represents a specific region with similar environmental conditions.



**Environmental input data**

ForClim contains a stochastic weather generator where the long-term averages and standard deviations of monthly mean temperature and log-transformed precipitation sums along with their cross-correlation serve as input (Bugmann, 1994; Risch

et al., 2005). These climatic input variables were derived from the CHELSA data set version 2 (Karger et al. 2017) with a horizontal resolution of 30 arc seconds. The plot's slope and aspect (represented as kSlAsp in the model; cf. Bugmann, 1994) are input variables as well; they were derived from the Copernicus digital elevation model EU-DEM (EU-DEM 2020) with a spatial resolution of 25 m, which was further processed with QGIS (QGIS Development Team 2022) to calculate slope and aspect on a spatial resolution of 100 m. The so-called "bucket size", i.e., the plant-available water storage capacity of the soil,

was derived with a Random Forest model trained with expert assessments of the soil quality of a subset of the plots (for details see Käber et al., 2023).

**The forest gap model ForClim**

ForClim is a dynamic vegetation model that simulates the processes of growth, mortality, and regeneration (often also called "establishment") of individual trees via species-specific size cohorts (Bugmann, 1994). ForClim classifies as a forest gap

model (Shugart, 1984) and simulates forests on independent patches, each of a size of 800 m². By default, 100 patches (i.e., 8 ha) comprise a forest stand to obtain realistic averages of forest dynamics across patches. The model uses an annual time step and represents trees as cohorts with the properties number of trees (Trs), their diameter at breast height (DBH), height, leaf area, and stress level. Here, we used two variants of the regeneration module within ForClim v4.0.1 (Huber et al., 2020).

**The two regeneration models**

The ForClim regeneration module initiates new cohorts of trees based on a) site variables for climate and soil in combination with b) species traits, and c) state variables of forest structure. The species traits of drought tolerance, temperature, and light requirements originate from Ellenberg (1986) (cf. latest translated edition, Leuschner and Ellenberg, 2017) and the FORECE model (Kienast, 1987). They define thresholds (so-called "establishment flags", EFs) that must be fulfilled for a species to qualify for establishment at a DBH of 1.27 cm. In this study, we focus on a simple and a more complex variant of the

regeneration model. Below, a brief summary of the two models is provided, followed by an explanation of the EFs investigated here. For more details, see Appendix B and the original documentation of the simple and complex model in Bugmann (1994) and Huber et al. (2020), respectively.

The simple model simulates tree regeneration for each species independently and corresponds to the original ForClim *establishment module* in Bugmann (1994), which is the same as model variant 1 in Huber et al. (2020). EFs in this model

indicate either "suitable" or "not suitable", i.e., they are binary. In a first step, the annual regeneration probability (kEstP), modulated by species-specific EFs, determines whether regeneration for each species takes place. Second, if regeneration of a species takes place, the potential maximum number of new trees for that species is calculated from (1) a regeneration intensity





parameter (kEstDens, which is the maximum tree establishment density per species $[m^{-2}yr^{-1}]$) and (2) the species-specific successional strategy (i.e., shade-intolerant species have a higher number of seeds and thus offspring compared to shade-

intolerant species). Third, the actual number of new trees per species is derived by drawing a random number between 1 and the potential maximum number of trees for each species.

The complex model includes a mechanism for competition and was first introduced as variant 11 in Huber et al. (2020). EFs in this model are continuous, which allows for a more nuanced gradient from "suitable" to "not suitable". In the first step, kEstP as modulated by a drought index and degree-days is used to determine if regeneration for any species takes place.

Second, if regeneration does take place, the total potential number of new trees over all species is calculated from (1) a regeneration intensity parameter (kTrMax, which is the absolute maximum number of trees $[ha^{-1}]$) and (2) a drought index, degree-days, and the continuous EFs. The actual number of new trees *over all species* is then derived by drawing a random number between 1 and the potential maximum number of trees over all species. Third, the number of new trees per species is calculated by multiplying the actual number of trees over all species by the species-specific ratio of each species' EF and the

sum over the EFs of all species.

**Establishment flags regarding light, temperature, and soil moisture**

In the present study, we focus on three of the five EFs that are used in the two models (Table B1). The definition of these three EFs (for light, drought, and degree-days) is given below.

The available light establishment flag (ALEF) evaluates whether the sunlight available at the forest floor ($gAL_0$, see p. 63 in

Bugmann (1994) for details) matches a parameter for species' light requirements to regenerate ($kL_{y,s}$). $kL_{y,s}$ is derived from indicator values regarding the light requirements of young trees ($L_s$) ranging from 1 to 9 (Leuschner and Ellenberg, 2017) with

$$kL_{y,s} = \begin{cases} 0.025 \cdot (L_s - 1) & L_s < 5 \\ 0.1 \cdot L_s - 0.4 & \text{else} \end{cases}. \qquad \textbf{eq. 1)}$$

For each species s, the binary EF ($ALEF_{b,s}$) in the simple model is calculated with

$$ALEF_{b,s} = \begin{cases} 0 & gAL_0 < kL_{y,s} \\ 1 & \text{else} \end{cases}, \qquad \textbf{eq. 2)}$$

while the continuous EF ($ALEF_{c,s}$) in the complex model is calculated with

$$ALEF_{c,s} = \begin{cases} 0 & gAL_0 < kL_{y,s} \\ 1 & gAL_0 \geq kL_{y,s} + 0.5 \\ \frac{gAL_0 - kL_{y,s}}{0.5} & kLy \leq gAL_0 < kL_{y,s} + 0.5 \end{cases}, \qquad \textbf{eq. 3)}$$

where the value 0.5 refers to the highest $kL_{y,s}$ (cf. eq. 1) and serves as a buffer for the transition from zero to one.

The degree-days establishment flag (DDEF) evaluates whether the annual degree-day sum (gDD, see p. 81 in Bugmann (1994) for details) matches the species' minimum degree-day requirement (kDDMin). The values of kDDMin originate from Kienast





(1987), who derived climatic variables from Müller (1982) and Rudloff (1981) for multiple geographic locations and elevations within the species' distribution range (Ellenberg and Klötzli, 1972; Meusel et al., 1965). This approach was further improved by applying a site-specific bias correction (Bugmann, 1994). Note that this parameter has never been modified to reflect possible deviations regarding the regeneration niche. In this study, we distinguish between the original parameter (kDDMin of adults) and $kDDMin_y$, which applies to the regeneration. For each species s, the binary EF ($DDEF_{b,s}$) in the simple model is calculated with

$$DDEF_{b,s} = \begin{cases} 0 & kDDMin_{y,s} \geq gDD \\ 1 & else \end{cases}, \qquad \text{eq. 4)}$$

while the continuous EF ($DDEF_{c,s}$) in the complex model is calculated with

$$DDEF_{c,s} = \begin{cases} 0 & kDDMin_{y,s} \geq gDD \\ 1 & gDD \geq kDDMin_{y,s} + 256 \\ \frac{gDD - kDDMin_{y,s}}{256} & kDDMin_{y,s} < gDD < kDDMin_{y,s} + 256 \end{cases}, \qquad \text{eq. 5)}$$

where the value 256 refers to the lowest $kDDMin_{y,s}$ and serves as a buffer for the transition from zero to one.

Lastly, the soil moisture establishment flag (SMEF) evaluates whether the drought index (gDr), defined as the ratio of actual evapotranspiration and water demand by the atmosphere (i.e., potential evapotranspiration) matches the species' threshold for this index, i.e., the drought tolerance (kDrTol). The original trait values for drought tolerance range from 1 to 5 ((Leuschner and Ellenberg, 2017) and were scaled between 0.06 and 0.3 (i.e., 30 %). The evolution of the formulation of the drought index is documented in Bugmann (1994); Bugmann and Cramer (1998); and Bugmann and Solomon (2000), including its integration in the regeneration model (Didion et al., 2009a). Similar to DDEF, the original parameter (kDrTol of adults) and $kDrTol_y$ are distinguished here, and the latter applies to regeneration only. For each species s, the binary EF ($SMEF_{b,s}$) in the simple model is calculated with

$$SMEF_{b,s} = \begin{cases} 0 & gDr > kDrTol_{y,s} \\ 1 & else \end{cases}, \qquad \text{eq. 6)}$$

while $SMEF_{c,s}$ in the complex model is calculated with

$$SMEF_{c,s} = \begin{cases} 0 & gDr > kDrTol_{y,s} \\ 1 & gDr \leq kDrTol_{y,s} - 0.08 \\ \frac{kDrTol_{y,s} - gDr}{0.08} & kDrTol_{y,s} > gDr > kDrTol_{y,s} - 0.08 \end{cases}, \qquad \text{eq. 7)}$$

where the constant of 0.08 indicates the lowest $kDrTol_{y,s}$ and serves as a buffer for the transition from zero to one, i.e., the EF being fulfilled or not (cf. Huber et al., 2020).

The two EFs in the model that are not considered here are the winter temperature establishment flag (WTEF), which depends on minimum tolerated winter temperature and chilling requirements (Bugmann, 1994; Bugmann and Cramer, 1998; Kienast,



1987); and the browsing pressure flag, which depends on the species' susceptibility to ungulate browsing (Didion et al., 2009b). WTEF is correlated with DDEF and excluded from the calibration to avoid too many degrees of freedom. We therefore used

the default parameterization for WTEF. Because no site information on browsing pressure was available, we decided against using this factor in the calibration. Instead, we kept browsing pressure constant across all sites at its default value of 20 %.

**Likelihood**

We estimated the three species-specific ($kL_{y,s}$, $kDDMin_{y,s}$, and $kDrTol_{y,s}$) and two general (kEstDens, kTrMax) regeneration parameters of ForClim based on the recruitment rates observed in the EuFoRIa reserves. The species parameters were estimated

for 11 out of 30 simulated species. The species not considered for calibration were simulated with their default parameters (cf. Huber et al., 2020). Tree recruitment was defined as the number of trees that pass an inventory-specific DBH threshold. Observed decadal tree recruitment rates $R_{i,s}$ were calculated for each plot i and species s with

$$R_{i,s} = \frac{\sum_{p=1}^{Nperiods} R_{i,s,p}}{T_i \times 10}, \qquad \text{eq. 8)}$$

where p is the inventory period and $T_i$ is the total number of years between the first and the last inventory at plot i. Simulated tree recruitment rates $\widehat{R}$ were calculated as

$$\widehat{R}_{i,s} = \frac{1}{Nrep_i}\sum_{k=1}^{Nrep_i} \frac{\sum_{p=1}^{Nperiods}\sum_{j=1}^{Npatch_{i,s,p,k}} nTrs_{i,s,p,k,j}}{T_i \times 10}, \qquad \text{eq. 9)}$$

where $nTrs_{i,s,p,k,j}$ is the number of recruited trees for one patch j, inventory period p, and repetition k. Each simulation during the calibration was conducted on ca. 100 patches (i.e., ca. 8 ha) to reduce the variability caused by the k stochastic realizations of the point process embodied in ForClim. The trees in the initial forest inventory were randomly distributed to each of the 100 patches (each with a size of 0.08 ha) proportionally to actual plot size until a full repetition exceeded 100 patches. If one repetition was not a multiple of the patch size of 0.08 ha, the difference in exceeded plot area determined the proportion of

additional trees drawn from all trees in the initial forest inventory to populate the patches. The number of repetitions $Nrep_i$ for each plot i emerges from the next higher integer to $\frac{8\,ha}{A_i}$, where $A_i$ is plot size in ha. This resulted in an average of 8 repetitions k across all sites, but with a range from plot sizes of 0.2 ha (k = 40) to two sites with plot size > 4 ha (k = 2). The number of patches j ($Npatch_{i,p,k}$) within one repetition k is the next higher integer to $\frac{100\,patches}{Nrep_i}$.

We used Bayesian inference to estimate the parameters of the two ForClim regeneration models and their uncertainties. We

assumed that the observations $R_{i,s}$ for site i and species s with the parameter vector θ P($R_{i,s}$ | θ) were negatively binomially distributed, leading to a likelihood per observation of



$$P\left( R_{i,s} \mid \theta \right) = \text{NegBinomial2}\left( R_{i,s} \mid \widehat{R}_{i,s}, \phi \right), \qquad \text{eq. 10)}$$

which includes the mean $\widehat{R}_{i,s}$ predicted by ForClim and the dispersion parameter $\phi$ of the negative binomial distribution, which can be interpreted as a measure of residual variation. The predicted mean $\widehat{R}_{i,s}$ is governed by the calibrated values for $kL_y$, $kDDMin_y$, $kDrTol_y$ and the respective regeneration intensity parameter (kEstDens & kTrMax) of the two models, as explained above.

We assumed that the dispersion may vary with species, DBH and plot size according to the formula

$$\phi = \phi_s \times e^{\phi_{DBH} \times DBH_i + \phi_A \times A_i}, \qquad \text{eq. 11)}$$

where $\phi_s$ is the species-specific dispersion, $\phi_{DBH}$ is the effect of the diameter threshold $DBH_i$ and $\phi_A$ the effect of plot size $A_i$ on the dispersion. We used an exponential function to only allow for positive values, as required by the negative binomial distribution.

The joint likelihood $P_s\left( y_s \mid \theta \right)$ for each species s is the sum of the log-likelihoods over all plots i and species s:

$$P\left( y \mid \theta \right) = \frac{\sum_{s=1}^{\text{Nspecies}} P_s\left( y_s \mid \theta \right)}{\text{Nspecies}} \qquad \textbf{eq. 12)}$$

To reduce the absolute magnitude of (stochastic) likelihood differences and implicitly also account for structural model error (see Oberpriller et al., 2021), we re-scaled the joint likelihood by a factor of 1/12 (i.e., Nspecies, which includes "other" species). The effect of this re-scaling is that uncertainties get wider and MCMC samplers are less prone to get stuck in local optima, while the optima retain their location.

In total, the likelihood depends on a vector $\theta$ consisting of 48 parameters, as we estimated three ecological threshold parameters ($kL_y$, $kDDMin_y$, and $kDrTol_y$; cf. Figure B1) and separate dispersion parameter $\phi_s$ for each of the eleven species, plus one extra dispersion parameter for all other species, two parameters for the effect of DBH and plot area ($\phi_{DBH}$ and $\phi_A$), and one parameter for the regeneration intensity (kEstDens or kTrMax) for the simple or the complex model, respectively. We defined wide uniform priors for each parameter that comprises the full range from the species' lowest and highest values in the default parameterization in ForClim (cf. Table B1).

**Model initialization**

To initiate the calibration runs, we had to resolve the issue that trees below the inventory-specific DBH threshold (i.e., small trees), which may have been present in reality, are obviously not contained in the data. Ignoring these trees in the initialization would create a temporal lag of tree recruitment (i.e., trees surpassing the DBH threshold), connected with a potentially





significant underrepresentation of tree regeneration directly after model initialization. To overcome this problem, we initialized unobserved trees below the DBH threshold with the model's steady state (i.e., equilibrium of regeneration).

This steady state was determined by running the simulation with the stand structure of the initial forest inventory for 50 years, suspending all processes affecting trees above or equal to the DBH threshold. During this "spin-up" phase, trees above the DBH threshold that are included in the initial forest inventory could neither die nor grow, but still modulated the variables of

stand structure that affect regeneration (i.e., $gAL_0$, Trs). In contrast, trees below the DBH threshold were allowed to grow and die under the conditions observed in the initial inventory. If these newly regenerated trees grew larger than the DBH threshold during the spin-up, they were removed. This means that these trees did not die from mechanisms that simulate tree mortality in the model but were forcefully removed from the simulation to avoid the accumulation of trees with a DBH close to the DBH threshold. Visual inspection of the simulation results showed that an equilibrium of the stand structure below the DBH

threshold was reached after approximately 50 simulation years, which was the reason to fix the spin-up to this time period. After the spin-up, the simulation was continued with all trees and running all model processes (i.e., regeneration, growth and mortality).

**Posterior estimation**

We calibrated the model using the differential evolution sampler (DEzs, ter Braak & Vrugt, 2008) as implemented in the R

Package *BayesianTools* (Hartig et al., 2019). We sampled with two independent sets of three chains (i.e., a total of six chains) for all 353 training plots. The z-matrix was re-initialized at the beginning of the sampling procedure (at 5000 to 6800 and 2000 to 3700 iterations for the simple model and the complex model, respectively) to improve the mixing of the chains. This was necessary because of the very wide prior range for the dispersion parameters, which led to a degenerated z-matrix. The same procedure was applied to improve mixing the chains after 120'000 to 139'300 (simple model) and 120'000 to 145'500

iterations (complex model). For the simple model, the upper prior range for the kEstDens parameter had to be adjusted from 0.02 to 0.2 after 139'300 iterations. Ultimately, after 191'600 (simple model) and 200'900 iterations (complex model), one set of three independent chains converged, as judged by the visual inspection of the chains and Gelman and Rubin's MCMC Convergence Diagnostic (cf. Table A4). The other set of independent chains did not fully converge, mostly because of one chain being stuck for the species-specific dispersion parameters. Computational constraints did not allow for running the

sampler even longer. One single simulation took 3 seconds per plot, i.e., 6 chains times 353 calibration plots resulted in a total computation time of 1.765 h per iteration without parallelization. Fortunately, the Euler High Performance Computing Cluster of ETH Zürich enabled us to use 1000 cores (500 per model variant) and sufficient memory. The effective computing time, including the overhead when utilizing all resources, was 10-15 seconds per iteration and ca. 25 days in total for each model variant.

The posterior distribution from the calibration consisted of 1'000 samples drawn from the last 32'300 (simple model) and 45'400 iterations (complex model). The simulations from the posterior parameter distribution provided posterior estimates of decadal tree recruitment rates ($\widehat{R}_{i,s}$) for all 343 test plots i and species s. The Mean Posterior Estimate (MPE) and the 80%





Credible Intervals (CIs) of $\widehat{R}_{i,s}$ from the 1'000 posterior simulations were used to assess residuals and evaluate model performance (see Table A4). The MPE and CIs for the parameter estimate of the posterior distribution were used to compare

the trait based model and the calibrated model.

**Performance comparison using Root mean squared error (RMSE) and marginal likelihood (ML)**

Model performance was assessed with the root mean squared error (RMSE) and the marginal likelihood (ML) on both training and hold-out data. The difference between the two metrics is that RMSE is a general metric of fit, while the marginal likelihood is Bayesian metric that relates to the Bayes Factor and posterior model weights and thus allows to compare the support for two

alternative models based on a specific likelihood.

For the simulations from the calibrated and trait-based models the RMSE was calculated for different DBH thresholds: for variable thresholds between plots and harmonized DBH thresholds of 7 and 10 cm. This harmonization was done by artificially increasing the DBH threshold in the observed and simulated data to mimic a consistent inventory design with a common DBH threshold. We calculated the RMSE from the training and test data based on the comparison of observed and simulated

recruitment across the species-unspecific, the species-specific, and as the average over the species-unspecific and species-specific RMSEs (Table 1).

The ML was calculated for the simulations from the calibrated models only because the likelihood relies on the dispersion parameters, which were not estimated for the trait based model. The ML is the average likelihood of the model given the training or the test data, averaged over the posterior parameter uncertainty (cf. Delpierre et al., 2019). We evaluated the

marginal likelihood in both cases on the validation data based on the posterior distribution inferred from the training data. This approach, which corresponds to the fractional Bayes factor (O'Hagan, 1995), avoids inconsistencies when comparing models with weak or uninformative priors. The Bayes Factor is then obtained by taking the ratio between two marginal likelihoods with $e^{M1-M2}$. This provides the relative posterior support of M1/M2 by the data (Kass and Raftery, 1995).

When interpreting the results, it is important to remember that both RMSE and ML as evaluated here will typically be higher

for more complex models on the training data, so the comparison of models with these metrics on the training data is of limited use. However, models can sensibly be compared by their performance on the hold-out, and it is also informative to look at the reduction of performance between training and hold-out, which gives an indication of overfitting.

**Posterior sensitivity**

Global sensitivity analyses allow for the assessment of model behavior across large parameter spaces. However, large

parameter spaces may also cover unrealistic parameter configurations, and computational requirements are high. Therefore, a strategy for constraining the parameter space to a relevant location is required (cf. Huber et al., 2018). We combined the benefits of a global and a local sensitivity analysis by constraining the parameter space via deriving a posterior distribution from the observations. This allowed us to evaluate model sensitivity with respect to the uncertainty derived from observed recruitment patterns in European forest reserves.



To analyze the sensitivity of regeneration to changes in the model parameters within the posterior distribution, we analyzed the effect of increased tolerance of trait values (i.e., lower $kL_y$, higher $kDrTol_y$, and lower $kDDMin_y$) on simulated recruitment within the posterior parameter range. This was done by modeling $\hat{R}$ with a GLM and a negative binomial distribution with z-scaled values of negative $kL_y$, negative $kDDMin_y$, and positive $kDrTol_y$ as predictors. This model was implemented using glmmTMB (Brooks et al., 2017).

**Results**

We first present the results on species traits and the regeneration intensity; second a comparison of model performance between the trait based and calibrated model variants; and third the sensitivity of species trait values within the posterior distribution.

**Species traits and regeneration intensity**

The trait-based regeneration niche differs from the regeneration niche that emerges from the model which was calibrated with the observations in unmanaged forest reserves following. Variation in these differences is evident between trait types, species, and model variants (Figure 1 and Table A2).

Light requirements stand out from the other traits because they were most sensitive during model calibration, as indicated by the narrow posterior distributions compared to the prior parameter range (Figure 1, left and Figura A2a). Most posterior estimates were not only narrow, but also supported by the trait values (Figure 2, left). Estimates of the complex model were generally closer to the trait values, with a good rank correlation (i.e., Spearman's rho = 0.57). Light requirements defined by traits for the simple model were systematically lower compared to the values emerging from the calibration. Nevertheless, a Spearman's rho of 0.94 indicates that the calibration put the species in a plausible order (Figure 2, top left). The estimates of the species-specific light requirements were more similar between both approaches for shade-tolerant tree species in general. Values from the calibration for *Quercus* spp., *Pinus sylvestris*, and *Betula* spp. were much lower compared to the trait values in the complex model but more similar in the simple model. For *Tilia cordata*, the calibration led to much higher light requirements compared to the trait based values with either model (cf. Figure 1, left).

Drought tolerance values from both approaches´ matched moderately well when using the complex model (Spearman's rho = 0.46) with relatively large CI, and the expectation was within the CI for eight species (Figure 1, center). The MPE from the calibration was close to the definitions of the trait values for five species. However, drought tolerance estimates from the calibration did not match the trait values well for the simple model (Spearman's rho = -0.18). Only for six species the wide CIs included the trait based values, and the MPE matched the trait values in the simple model for two species only. For *Quercus spp.* (complex model), *Tilia cordata* (simple model) and *Pinus sylvestris* (both models), the calibration led to much lower values compared to the trait values. Conversely, for *Alnus glutinosa* and *Fraxinus excelsior* the calibration resulted in higher values compared to the trait-based values (cf. Figure 1, center).





Estimates for the minimum degree-days had wide CI for both models and a low rank correlation between the approaches (Spearman's rho = 0.19 for the complex model and -0.3 for the simple model; Figure 1, right). Although the posterior CI of the calibration included the trait-based values for many species, the intervals were wide and the MPE values were close to the trait-based values only for two and three species in the complex and simple model, respectively. This indicates that the calibration values neither fully disagree nor perfectly match the trait-based values. For *Tilia cordata* (both models) and

*Carpinus betulus* (simple model), the calibrated values were much lower compared to the trait-based values. Conversely, the calibrated degree-day values for *Pinus sylvestris* (simple model) and *Abies alba* (both models) were considerably lower compared to the trait-based values (cf. Figure 1, right).

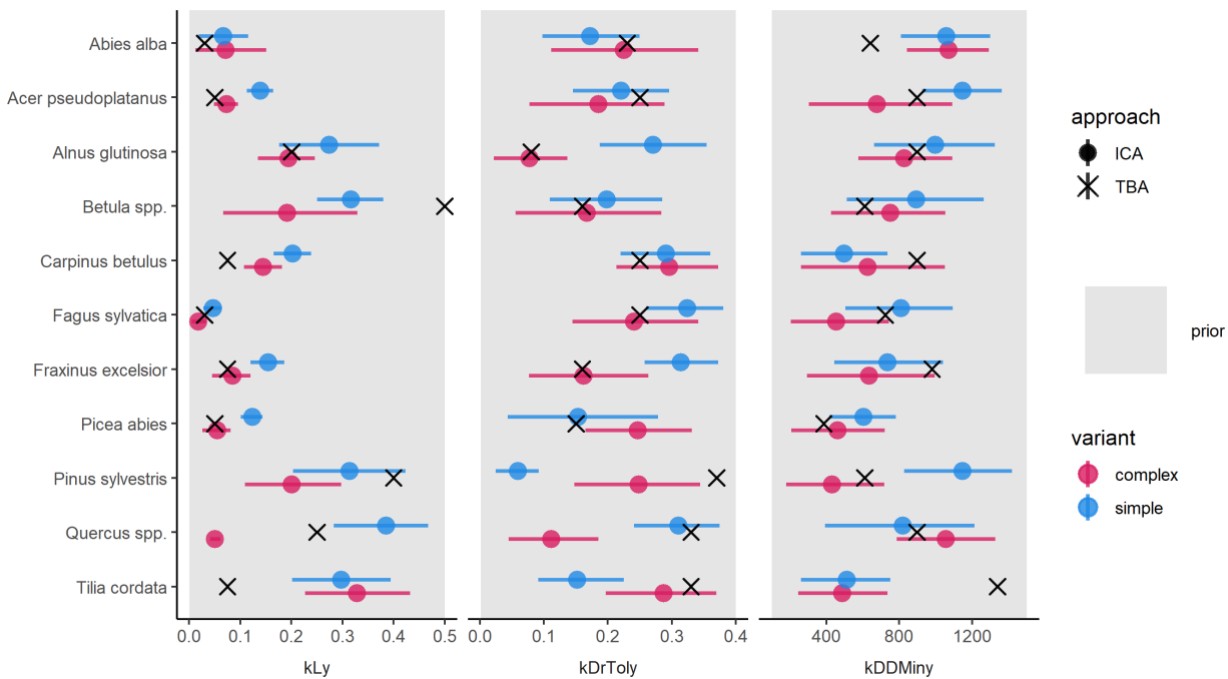

**Figure 1: Mean Posterior Estimate including the 80 % CI of the species-specific parameters for the complex (red) and the simple**
**model (blue). Point type indicates the inverse calibration approach (ICA) and the trait-based approach (TBA). The panels show the species trait values light requirements ($kL_y$), drought tolerance ($kDrTol_y$), and minimum degree-days ($kDDMinkL_y$). The prior parameter range is indicated by the grey rectangle in the background.**



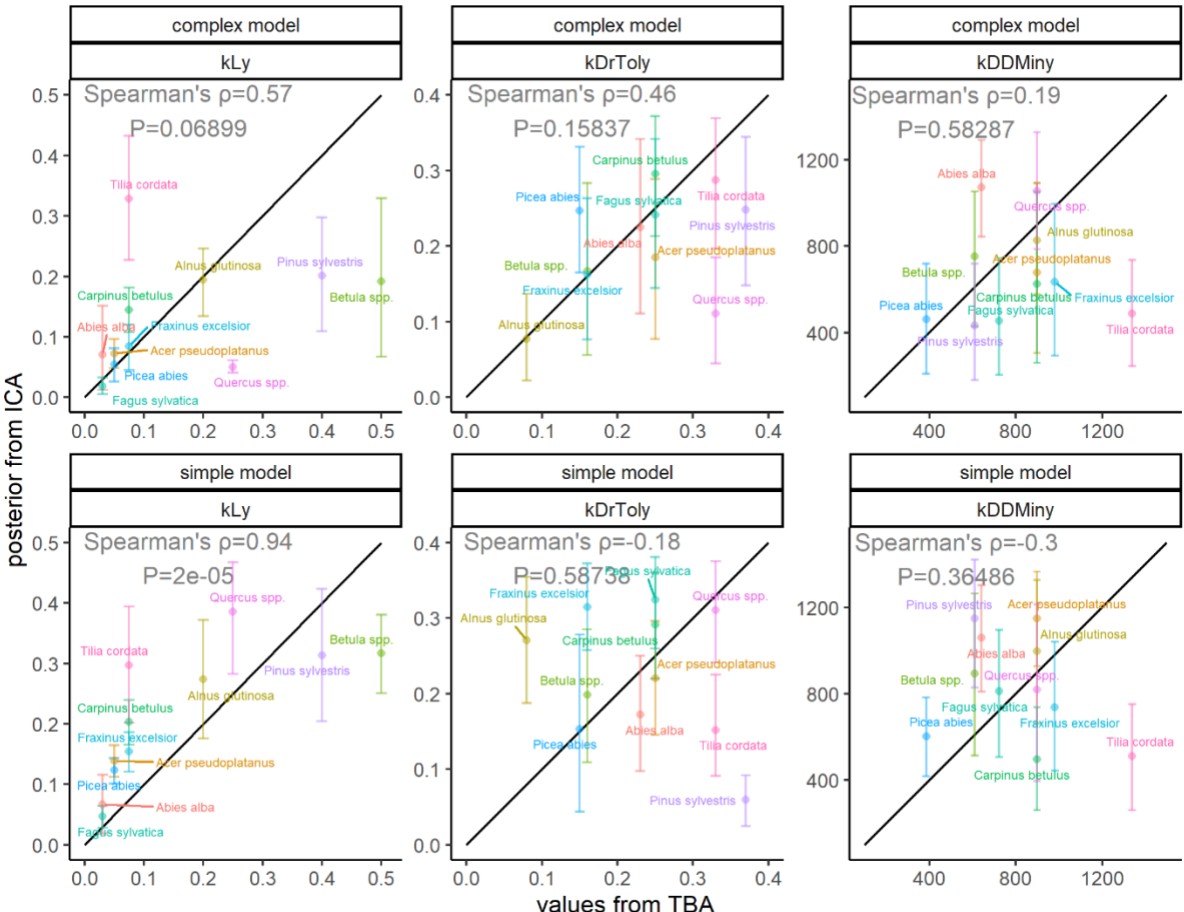

**Figure 2: Comparison of expected values for species-specific establishment thresholds based on ecological knowledge (TBA, trait-based approach; cf. Leuschner and Ellenberg, 2017) and MPE (ICA, inverse calibration approach, this study). The range displays the 80 % CI for the complex (top panels) and the simple model (bottom panels). The 1:1 relationship is indicated by the black line. Spearman's rank correlation and the p-values of the MPE and the trait-based approach (TBA) values are shown in each panel.**

The estimates of the calibration for general regeneration intensity were narrow compared to the prior (Figure 3) and therefor considerably sensitive. While the species-unspecific parameter kTrMax from the calibration (complex model) was significantly lower than its default value of the trait-based model, the species-specific parameter kEstDens (simple) was higher than its default value of the trait-based model (Table A1). Considering the interaction between kEstP (which was reduced by a factor of 1/5) and kEstDens or kTrMax (cf. model description in Appendix B), the overall amount of regeneration is generally lower for the calibrated model compared to the trait-based model. Specifically, for the complex model the MPE (kTrMax = 8762) suggests 25 times less maximum regeneration compared to the default values of the trait-based models (kTrMax = 50000 · 5 = 250000); and for the simple model the MPA (kEstDens = 0.022) suggests a reduction of ca. 24 % per species compared to the default trait-based model (kEstDens = 0.006 · 5 = 0.03). However, it is noteworthy that these parameters modulate regeneration differently and the magnitude of the deviation between the parameters in the calibrated model, the default trait-





based model, and the model variants is not directly translated into the simulated regeneration amount within the model (cf. Figure B1 and the full set of equations in Appendix B).

The coefficient for the effect of the DBH threshold in the species-specific dispersion was significantly different from zero, with an MPE of -0.39 and -0.41 for the complex and simple model, respectively. This indicates that dispersion increases for higher DBH thresholds (Figure 3c). The coefficient for the effect of plot size is slightly negative but not significantly different from zero (Figure 3d), which indicates that there is no significant effect of plot size on dispersion. However, a very weak positive effect of larger plots on dispersion is visible. On the species level, dispersion effects differed significantly, with the

lowest dispersion for *Fagus sylvatica* and the highest for *Quercus* spp. (Figure A1 and Table A3). These findings emerged from both the simple and the complex model (cf. Figure 3 c and d).

**Figure 3: Posterior distribution of the non-species specific parameters determining the amount of regeneration: a) kTrMax (complex model, red) and b) kEstDens (simple model, blue). Effects on the dispersion parameter $\phi$: c) DBH threshold and d) plot size. The**
**prior parameter in a) and b) is given by the extent of the graph. The prior range for the dispersion parameters was -5 to 5 and is not**



shown. Note that lower values of the dispersion parameter indicate higher dispersion. Consequently, negative estimates for dispersion are positive effects on the actual dispersion. Species' dispersion parameters are presented in Table A3.

## Model performance

The calibration led to somewhat better performance compared to the trait-based approach (Figure 4 and Table 1). Both model
variants performed better when calibrated and revealed the uncertainty of the posterior simulations. However, performance differed strongly between species. Most gains in performance coupled with a high degree of uncertainty were evident for *Abies alba* and *Tilia cordata* with both models (Figure 4). No increase in performance was evident for *Quercus* spp. and *Alnus glutinosa*, although high uncertainty of the simulations was evident for *Alnus glutinosa* with the complex model. Slight but distinct gains in performance were found for *Fagus sylvatica* and *Picea abies* with the complex model. Note that not only the
intercept for the comparison of observations and simulations changed, but also the slope, which indicates that this was not only due to the regeneration intensity parameters. Overall performance increased distinctly for the complex model, and considerably for the simple model (cf. RMSE in Table 1). In summary, the calibration clearly improved model performance for almost all species in the complex model and to a limited extent in the simple model. In addition, the uncertainty based on the posterior parameter distribution was clearly visible in the simulations. The RMSE decreased with increasing DBH threshold (Figure
A3) for both models.

The ML confirmed the higher performance of the complex model (ML = -391.61) compared to the simple model (ML = -401.86), which was also evident from the RMSE (Table 1 and Figure A2b). According to Kass and Raftery (1995), the Bayes Factor from the training data being above 200 suggested strong support for the complex model (Bayes Factor = 28282.54).

**Table 1: RMSE of simulated and predicted R and likelihood for the different models and approaches: inverse calibration approach**
**(ICA); trait-based approach (TBA). The marginal likelihood (ML) was derived from the posterior parameter distribution of the training data (N=353), and the test data (N=343). Species specific RMSE values are presented in Figure A4.**

|  | complex | | | | simple | | | |
|---|---|---|---|---|---|---|---|---|
|  | ICA | | TBA | | ICA | | TBA | |
|  | test | training | test | training | test | training | test | training |
| RMSE | 5.84 | 6.19 | 8.86 | 9.29 | 6.32 | 6.63 | 6.17 | 6.47 |
| ML | -391.61 | -411.38 | | | -401.86 | -419.65 | | |





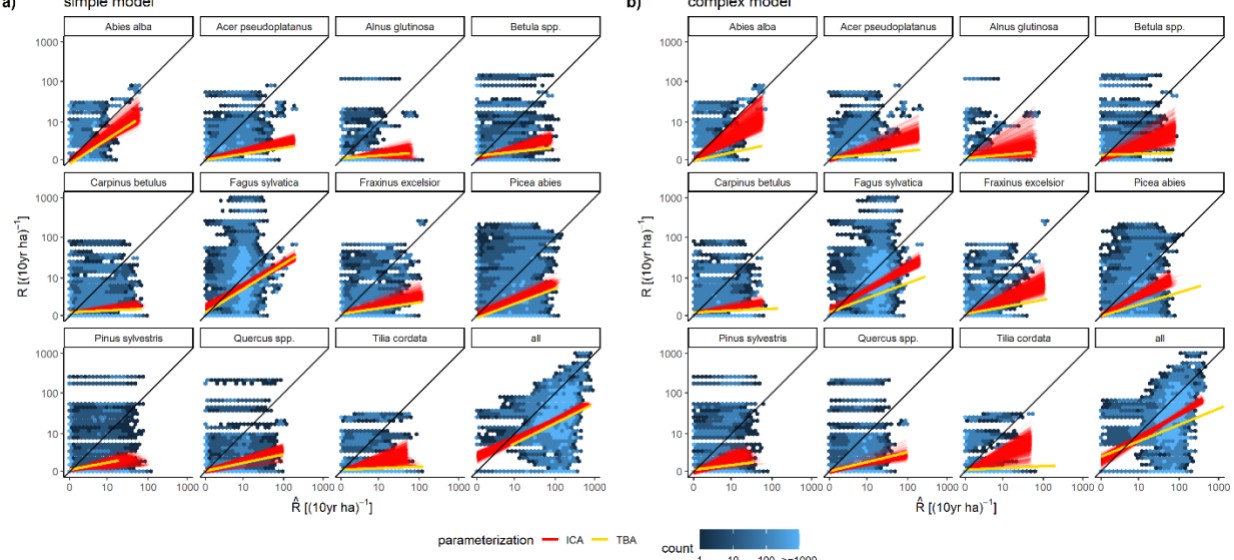

**Figure 4: Simulated vs. observed recruitment rates at 7 cm DBH of the 11 species for (a) the simple and (b) the complex model. The**
**red lines show linear regressions from 1000 simulated recruitment rates using the posterior parameter distribution at the 343 test**
**plots. The data points used for the regression are indicated by the blue color of the hexagons, where light blue indicates fewer points**
**and dark blue indicates more points. The yellow lines are characterizing the recruitment rates for the same test plots based on the**
**default parameter setting (TBA, trait-based approach).**

### Posterior sensitivity

The variation of the parameter values characterizing the species' light requirements had the strongest effect on conspecific

regeneration for either model variant (Figure 5 a and b, respectively). The variation of drought tolerance within the posterior,

which was rather high, had a much weaker effect on conspecific recruitment, and the minimum degree-days had a very low

effect on recruitment within the posterior parameter range. Interestingly, positive effects for heterospecific recruitment were

evident in the case of higher shade tolerance of *Quercus* spp. and *Acer pseudoplatanus*, but only in the complex model (Figure

5b). *Picea abies* (simple model) and *Fagus sylvatica* (complex model) showed the strongest negative effects on the emergence

of heterospecific recruitment (Figure 5).



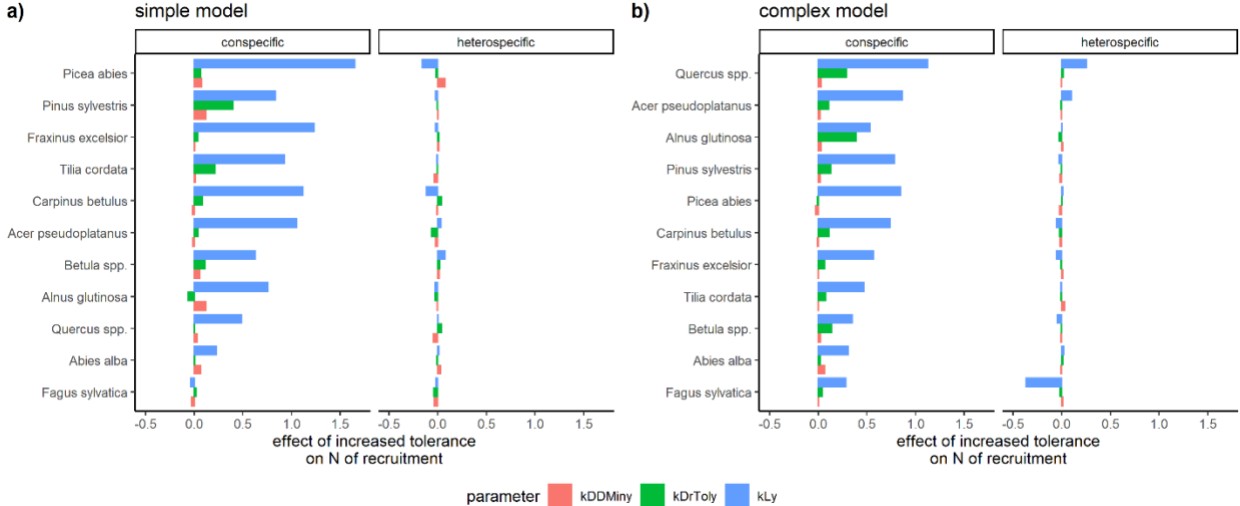

**Figure 5: Estimates of the effect of increased tolerances for $kL_y$, $kDrToly_y$, and $kDDMiny_y$ on the emergence of conspecific and heterospecific recruitment for the simple (a) and the complex model (b). The effect sizes correspond to the coefficients from a GLM that predicts recruitment with scaled tolerance parameters so that they have a mean of 0 and a standard deviation of 2; an increase of the value always indicates an improvement for the species.**

## Discussion

Below, we discuss the research questions with respect to the results and the ecological implications of our findings. First, we focus on the differences of the species-specific traits between the calibrated and the trait-based model; second, we evaluate how the structure of the simple and complex model affected performance; and third, we discuss technical advances and methodological aspects.

## Species traits

The species trait values varied considerably between the calibrated models, the trait based-models, and the two model variants. We aim to explain these differences by reflecting on a) the theoretical expectations of the two approaches, and b) the structural differences between the model variants.

Differences between the trait-based and calibrated models can be expected based on ecological considerations regarding the regeneration niche (Grubb, 1977) and ontogenetic niche shifts (Werner and Gilliam, 1984), as well as methodological aspects such as the importance of context for modeling trait-demography relationships (Yang et al., 2018). Specifically, we expected high sensitivity and a good match for the shade tolerance of the species because light availability is a key determinant of tree regeneration on small spatial scales (Collins and Good, 1987), and its context is modeled explicitly and in rather high detail in ForClim (cf. the direct link between stand structure and light availability in equations eq. 1 to eq. 3). In contrast, the context of trait values related to climate (drought tolerance and temperature requirements) is only vaguely defined by species distribution limits (Meusel et al., 1965) along macroclimatic gradients (Rudloff, 1981). In addition, the traits for light





requirements are differentiated between juveniles and adults (Leuschner and Ellenberg, 2017), while those related to climate

are not. Therefore, we expected less agreement between the calibrated and trait-based models for these latter drivers. Our results supported these expectations, as shown by the mostly good agreement between the light requirement parameters compared to climatic parameters in the two approaches. Notably, this pattern was found in both the simple and the complex model, and it thus appears to be a robust feature irrespective of the structure of the regeneration model. Furthermore, the drought-related parameters matched better between the calibrated and the trait-based models in the complex variant, which

indicates that trait values embedded in a model that connects drought effects and competition during regeneration have more support by the data. This interlink of competition and drought has also been demonstrated in grassland communities (Grant et al., 2014; Levine et al., 2022) and tree species mixtures (Jucker et al., 2014; Grossiord, 2020; Young et al., 2017; Clark et al., 2016; Ruiz-Benito et al., 2013; Haberstroh and Werner, 2022).

**Light**

The nuanced differences between the model variants in terms of the estimates of the species' light requirements can be put in context with the regeneration intensity parameter. The calibrated trait values from the complex model were almost identical for most species to the trait-based values (Larcher, 1996; Lyr et al., 1992), whereas in the simple model the light requirements were systematically lower in the calibrated compared to the trait-based models. This indicates that in the simple model, excessive recruitment levels (as embodied in the parameter kEstDens) were compensated for by erroneous light requirements

(cf. Figure 1). These inconsistencies may arise from the structure of the simple model, where the amount of recruitment is equal for all species that regenerate. Thus, the simple model lacks flexibility to a) generate an appropriate number of recruits for the dominant species and b) lower the number of recruits for less dominant species. This explanation is supported by two other findings: the estimates of light requirements for the often-dominant species *Fagus sylvatica* matched the trait-based values, while almost all other species had exaggerated estimates (Figure 1); and the sensitivity of *Fagus sylvatica* to light

within the posterior distribution was close to zero, with considerable uncertainty regarding the modulating effects of light for other species (Figure 5). The light-demanding tree species *Quercus* spp., *Pinus sylvestris* and *Betula* spp. along with *Tilia cordata* did not match expectations either, which is in line with this pattern. These findings suggest that structural problems regarding competition for light in regeneration models of European forests can be exposed by the behavior of *Fagus sylvatica*. Thus, if the competitive dominance of *Fagus sylvatica* is not captured appropriately, a calibrated model is likely to compensate

this elsewhere.

In contrast to the absolute values for light requirements, their ranking of the species was more similar between the approaches for the simple model (Figure 2). The lower rank correlation of the complex model is mostly due to much lower estimates for light demanding tree species such as *Betula* spp., *Pinus sylvestris*, and *Quercus* spp., thus suggesting that the simple model performs better in simulating regeneration of early successional species. One possible reason for this behavior could be that

all establishment factors (i.e., environmental drivers of regeneration) are assumed to be equally important in virtually all vegetation models, including ForClim (cf. Bugmann & Seidl 2022). This assumption has different implications for a model with competition between species (complex model) compared to a model without competition (simple model). This becomes




clear if we consider a stand with high light availability. Then, the light requirements of all species are fulfilled. Within the simple model, the species establishment count takes into account their successional strategy, while the complex model lacks

this mechanism and may favor species with higher suitability derived from factors other than light, thus blurring the overwhelming effect of the life-history strategies on the amount of tree regeneration under high light conditions (cf. Welden and Slauson, 1986). Subsequently, the simple model adjusts excessive regeneration by the processes of growth and mortality. By contrast, in the complex model there may be too few early successional trees, and subsequent compensation is insufficient. This notion is supported by the fact that RMSE decreased with higher DBH thresholds and suggests that unrealistic

regeneration patterns must be compensated in vegetation models by subsequent growth and mortality (cf. Diaz-Yanez et al. in review).

In summary, the light niche of most species was recovered considerably well. If the identified inconsistencies regarding light requirements are caused by structural problems of the model, our results provide strong support for the quantification and ranking of species' trait based light requirements (i.e., the original parameterization of species light requirements in ForClim).

**Drought**

The estimates for drought tolerance were wide in the calibration for both model variants, and for the complex model the rank correlation between calibrated and trait-based values was better. The ranking of species trait-based drought tolerance values is ecologically plausible and widely accepted (Huber et al., 2020; Bugmann, 1994; Leuschner and Ellenberg, 2017). Yet, a key difference between the structure of the simple and complex model is competition during regeneration, which may explain the

better rank correlation for the complex model (cf., Grant et al., 2014; Andivia et al., 2018; Käber et al., 2023). However, the credible intervals of the calibration estimates were high, and various mismatches were evident. We surmise that they arise from and oversimplified representation of drought where nuanced differences between species drought tolerances and potential facilitation effects are not reflected (Lortie and Callaway, 2006). A different and more detailed perspective on modeling competition for drought is considering the intra- or interannual variability of water availability in contrast do species

phenological requirements (cf. Detto et al., 2022 and Levine et al., 2022). However, mismatches in temporal and spatial scales between the representation of drought in the simulations and actual drought conditions at the observed sites, coupled with errors in the input variables (climate and soil properties), and observations are possible reasons for high CIs (cf. Shoemaker et al., 2020). Consequently, we would expect higher predictive ability of tree species traits for drought on smaller scales with clearly defined relations between environmental drivers and outcomes, as shown by Li et al. (2022), who found that species

traits explained more variation in tree seedling performance under controlled conditions in experiments compared to large-scale studies (cf. Paine et al., 2015). For the simple model, the estimates did not follow a clear pattern, and it is difficult to assess whether the estimates that are close to the expectation (e.g., for *Picea abies*) are actually providing signal, or are just random. In spite of these uncertainties, it is noteworthy that some species-specific trait-based values of the were recovered with the calibration, thus providing at least some support by the calibrated drought-related regeneration niche as it was defined

by the traits.

**Temperature**



In contrast to the two other autecological parameters, the calibration estimates for the minimum degree-day requirements rarely matched the trait-based values. In general, the species minimum degree-days had wide CIs. We consider three factors to explain this. First, the manifold effects of temperature on regeneration at different scales, which cannot reasonably be aggregated into one single parameter, coupled with the fact that the original source of the trait values did not differentiate juvenile distribution ranges (Meusel et al., 1965; Kienast, 1987; Rudloff, 1981). Second, ontogenetic shifts (e.g., Vitasse, 2013) and demographic dependencies, i.e., the cumulated survival probability and growth over a trees lifetime (cf., Grubb, 1977; Heiland et al., 2022). And third, it is likely that temperature-related processes are limiting regeneration much less often in our data set compared to the persistent and strongly varying competition for light (cf. Grime and Mackey, 2002; Vincent and Harja, 2008). Distinguishing between filters for the macroclimatic factors along with dynamic small-scale filters might be a better conceptual basis for more realistic and more accurate tree regeneration models (but see Thakur and Wright, 2017). Consequently, with respect to process formulations in dynamic models, valid growth and mortality formulations might be more important for temperature-regeneration relations than the formulation of the initiation phase of tree regeneration.

**Model performance**

Generally, the calibration resulted in much better performance for the complex and a moderate improvement for the simple model compared to the trait-based approach. This is consistent with previous studies using Bayesian calibration of dynamic models (Augustynczik et al., 2017; Cailleret et al., 2019; Trotsiuk et al., 2020; Van Oijen et al., 2005). The main reason for this improvement in both model variants is the overall lower regeneration amount, which results from the combination of establishment probability and regeneration intensity. Thus, our results suggest that calibration can help to sharpen the estimates of regeneration parameters that are not well-constrained by standard empirical data.

The somewhat better performance of the complex model is best explained by the way the species-specific amount of regeneration is determined. While the simple model does this uniformly, the complex model distributes the regeneration to the species according to their environmental suitability. This is also reflected in the more realistic estimates of the regeneration niche along the drought gradient. Overall, our findings corroborate the considerations of Huber et al. (2020), who suggested the simultaneous use of different model variants. In our study, the regeneration patterns across the very heterogeneous forest types in our data set was captured much better by the complex model, which implicitly allows for differentiating processes (captured via the EFs) in the regeneration layer. In contrast, the ideas underlying the simple regeneration model, which was originally developed for multi-species forests with high evenness (Botkin et al., 1972), turned out to be less suitable for reproducing the observed regeneration patterns.

In addition to established performance measures such as RMSE or the Bayes Factor, the comparison of the default trait values and inversely calibrated trait values allowed us to evaluate whether the calibrated parameters are just 'degrees of freedom' that are used to make the model fit better to the data, or whether their estimates are plausible from an ecological perspective (cf. Hellegers et al., 2020). Based on the discussion on species trait values above, this leads to the conclusion that the simple model is more realistic for the factor light, whereas the complex model captures processes related to drought better.





### Methodological considerations

#### Spin-up phase

We used a spin-up phase for dealing with the lack of information on small trees (i.e., the trees that are smaller than the diameter threshold, which inevitably has to be used in any inventory) in the initial state of the forest inventory. The spin-up phase proved to be a good solution because regeneration amounts were generally in agreement with observations. However, we were unable to evaluate whether the assumption of a steady state of regeneration below the DBH threshold was realistic. For this, data with much higher temporal resolution and a low DBH threshold ($\leq 1.27$ cm) would be necessary. Theoretically, our approach would lead to biased regeneration if actual conditions for regeneration are significantly different from the conditions observed in the initial inventory. For example, lower light availability in the actual conditions would lead to a bias towards shade intolerant species, conversely higher light availability would lead to a bias towards shade tolerant species. In addition, the overall regeneration amount could be affected by these biases. Thus, we encourage future studies to test the implications of our assumptions to evaluate potential bias introduced by our approach.

#### Dispersion

Processes that are not considered in the models could explain further variation of parameters and performance between approaches and model variants. We found that the dispersion parameter of the negative binomial distribution was mostly determined by ecological processes: large differences of dispersion between species indicate that species-specific factors play a key role, as discussed below.

One such factor is the regeneration strategy, for which light requirements usually are a good predictor (Grime, 1977). Species with high light requirements that require disturbances for regeneration (e.g., *Betula* spp., *Pinus sylvestris*) featured higher dispersion than typical late-successional, shade-tolerant species (e.g., *Fagus sylvatica* or *Picea abies*. This pattern was also reflected for intermediate species on a gradient from low to high light requirements. However, not all species follow this pattern.

Migration limitations are another factor that are likely to contribute to species-specific range limits. Specifically, the range limits of *Abies alba*, *Carpinus betulus*, and *Quercus* spp. are potentially determined by lags in postglacial range expansion (Mauri et al., 2022; Svenning et al., 2008) and its interplay with long-term demographic processes and competition (Scherrer et al., 2020). The mismatch between estimated and ecologically plausible parameters could be caused by the model assumption that seeds of all species are available all the time, and the associated absence of dispersal limitations in the model. Dispersion parameters that are based on real-world observations account for such problems when using likelihood-based approaches for model evaluation. Consequently, species-specific clustering (e.g., random draws from a negative binomial distribution) could substitute mechanisms that are not explicitly included in dynamic forest models. However, the parameterization of such mechanisms would be challenging because it would require a process-based justification, otherwise dispersion parameters are only useful as a statistical measure for clustering in observed data (Hartig et al., 2012).





Overall, whether the incorporation of dispersion is beneficial in a model calibration study depends on the purpose of the study. For achieving higher accuracy of stand-level predictions, our study demonstrates that dispersion must be accommodated. Especially validation and calibration studies require dispersion components to enable a reliable comparison of simulations with observations of tree regeneration. From a theoretical point of view, however, the incorporation of dispersion is not necessarily required. For example, a study on different management scenarios without considering dispersion can still generate valuable insights for silvicultural decisions if the assumptions and context are clearly defined.

**Likelihood**

The likelihood proved to be generally useful for our model calibration. Nevertheless, several aspects regarding the approach applied here can be improved in follow-up research. First, we deal with a stochastic likelihood that makes it extremely difficult for the DEzs sampler to efficiently sample the parameter space. We acknowledge that theoretically other approaches such as Bayesian Synthetic Likelihood (Wood, 2010) or Approximate Bayesian Computation (Csilléry et al., 2010) might solve the issue of intractable likelihoods more elegantly than our approach. However, computation time would be a major challenge if one wanted to apply these alternative approaches. Second, we focused on decadal average tree recruitment rates as a benchmark for evaluating the tree regeneration niche. This aggregates over many subprocesses and does not explicitly include the factor time in the likelihood. We did not consider early growth or mortality just after establishment either. Future studies may consider all three demographic processes simultaneously to construct an improved benchmark of model accuracy (Bröcker and Smith, 2007; Dietze, 2017). Third, our study covered only very few boreal plots, and rarely the transition towards very dry, Mediterranean-type forest ecosystems. Thus, future studies could benefit strongly from extending the environmental gradients to more extreme climates, so as to reduce parameter uncertainty. Thus, our study also underlines the importance of long-term monitoring of forest ecosystems over a wide range of conditions (cf. Hanbury-Brown et al., 2022).

**Conclusion**

This study aimed to compare two tree regeneration models of different complexity and to examine their ability in capturing the regeneration niche of eleven tree species in unmanaged European forests. Furthermore, we sought to gain a deeper understanding of the effectiveness of two approaches to parameterize tree regeneration in dynamic forest models.

The comparison of the regeneration niche emerging from the inverse calibration approach and the predefined niche of the trait-based approach revealed that calibration led to better predictions of tree regeneration. The improvements were mostly caused by the lower regeneration intensity compared to the trait-based models. Decreases in regeneration intensity were modulated by competition for light with a subordinate role of drought. Temperature was not sensitive, and based on the EuFoRIa dataset it was not possible to recover the temperature-based niche. The mismatches between predefined and inversely calibrated trait values led to the conclusion that competition for light is the key processes for tree regeneration along with parameters that modulate tree regeneration amount. We therefore hypothesize that climatic drivers must become more important after initial establishment., having pronounced effects on tree growth and, indirectly, on mortality.



Furthermore, we found that a more complex model that incorporates competition during regeneration features a higher
performance compared to a simple model without competition. This highlights the importance of considering the interactions
between species during the regeneration process and underscores the potential of adding model complexity for improving
model performance.

Future research faces the challenge of identifying the sweet spot between simulating realistic, nuanced regeneration amounts
for individual species on the one hand and excessive regeneration that must be regulated later in tree life by growth and
mortality on the other hand. While the former might expose more structural problems of the model with the consequence of
unrealistic species composition and insufficient regeneration intensity, the latter potentially results in overoptimistic
predictions of forests regenerative capabilities, with consequences e.g. for the assessment of the adaptive capacity of forests
to climate change.

Overall, we encourage the use of inverse calibration to improve the understanding of the relation of real-world observations
and tree regeneration models. Our major contribution to improve tree regeneration models lies in the finding that overall
regeneration intensity and light availability are the most important factors that govern tree regeneration. Conversely,
macroclimatic drivers (i.e., effects of climate) are not expected to directly alter the emergence of small trees but rather affect
tree regeneration by modulating the light availability via increased mortality of larger trees. Thus, the accuracy of predictions
of tree regeneration for the resilience of forests under climate change may depend more strongly on the representation of
within-stand dynamics than the species range limits along large climatic gradients.

**Code and data availability**

The current version of *model* is available from the project website: https://ites-fe.ethz.ch/openaccess/products/forclim under
the GNU GENERAL PUBLIC LICENSE v3. The exact version of ForClim used to produce the results used in this paper is
archived on Zenodo (https://doi.org/10.5281/zenodo.8334092) (Käber et al., 2023b), as are input data and scripts to run the
model and produce the plots for all the simulations presented in this paper (https://doi.org/10.5281/zenodo.8334092) (Käber
et al., 2023b).

**Acknowledgements**

We thank all researchers of the European Forest Reserves Initiative (EuFoRIa, www.euforia-project.org) who contributed
long-term data for this study. EuFoRIa has been invaluable for this study, and we are grateful for their support and trust. In
particular we thank Thomas A. Nagel (University of Ljubljana, Ljubljana, Slovenia), Tuomas Aakala (University of Eastern
Finland, Joensuu, Finland), Markus Blaschke (Bavarian State Institute for Forestry, Freising, Germany), Bogdan Brzeziecki
(Warsaw University of Life Sciences, Warszawa, Poland), Marco Carrer (University of Padova, Legnaro, Italy), Eugenie
Cateau (Reserves Naturelles de France, Quetigny, France), Georg Frank (Austrian Federal Research Centre for Forests, Natural





Hazards and Landscape (BFW), Wien, Austria), Shawn Fraver (School of Forest Resources, University of Maine, Orono,
Maine, USA), Jan Holik (Silva Tarouca Research Institute, Brno, Czech Republic), Stanislav Kucbel (Department of
Silviculture, Faculty of Forestry, Technical University Zvolen, Slovakia), Anja Leyman (Research Institute for Nature and
Forest, Brussels, Belgium), Peter Meyer (Northwest German Forest Research Institute, Göttingen, Germany), Renzo Motta
(University of Torino, Torino, Italy), Pavel Samonil (Silva Tarouca Research Institute, Brno, Czech Republic), Lucia Seebach
(Forest Research Institute of Baden-Württemberg, Freiburg, Germany), Jonas Stillhard (Swiss Federal Institute for Forest,
Snow and Landscape Research, Birmensdorf, Switzerland), Miroslav Svoboda (Czech University of Life Sciences, Prague,
Czech Republic), Jerzy Szwagrzyk (University of Agriculture, Krakow, Poland), Kris Vandekerkhove (Research Institute for
Nature and Forest, Brussels, Belgium), Ondrej Vostarek (Czech University of Life Sciences, Prague, Czech Republic), Kamil
Kral (Czech University of Life Sciences, Prague, Czech Republic), Tzvetan Zlatanov (Institute of Biodiversity and Ecosystem
Research, Bulgarian Academy of Sciences, Sofia, Bulgaria). In addition, we thank Hussain Abbas who provided programming
support and facilitated the use of the Euler HPC cluster. This study is part of the PhD project by Yannek Käber and was funded
by ETH Zurich (Grant ETH-35 18-1).

**Author contribution**

YK, FH, and HB conceived the idea, developed the concept of this study; YK led the writing of the paper, conducted the
analysis, and created the graphics. FH provided statistical support. HB secured funding. All authors contributed to paper
writing.

**Competing interests**

The authors declare that they have no conflict of interest.

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





# Appendix A

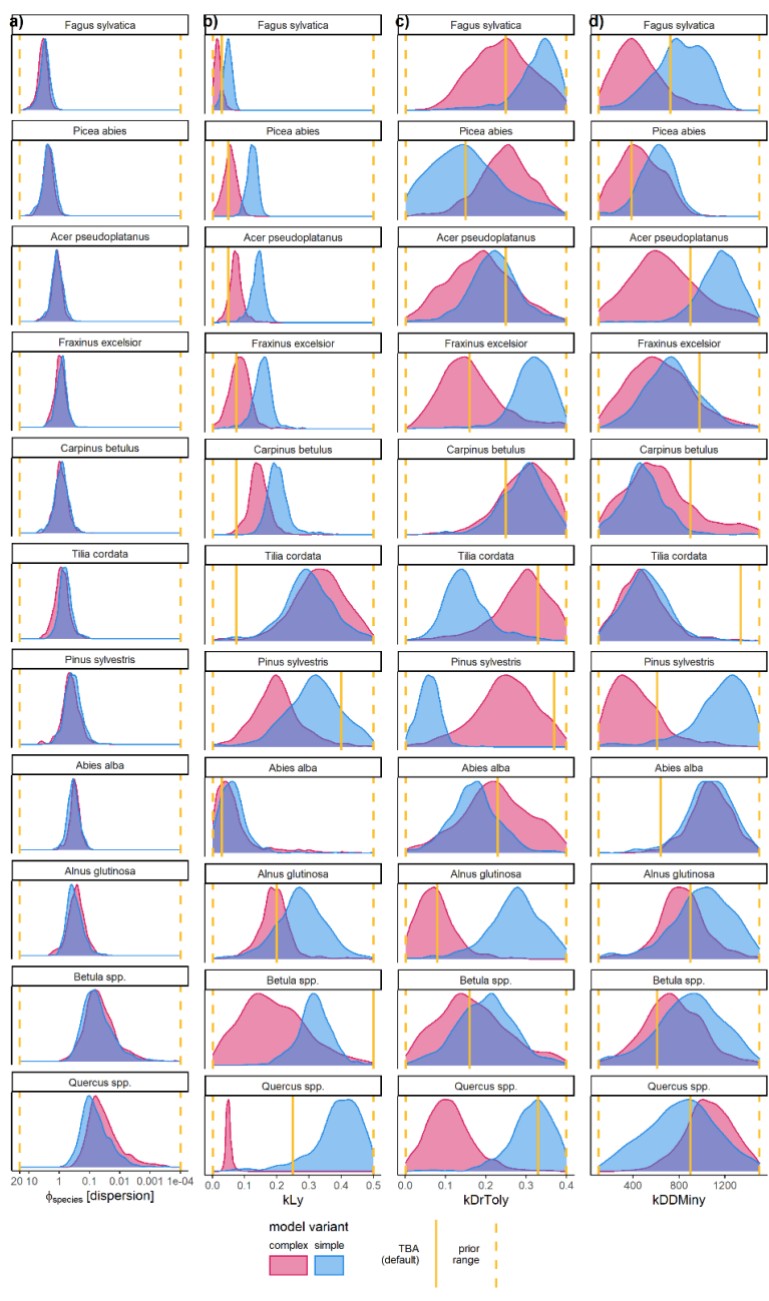

**Figure A1: Posterior distribution of species-specific parameters for the complex model (red) and the simple model (blue). The first column a) shows the species-specific dispersion parameter $\phi$. The other columns show the ecological regeneration thresholds: b) light requirements (kLy), c) drought tolerance (kDrToly), and d) degree-days (kDDMiny). The species are sorted row-wise according to their average dispersion across both models. The prior parameter range and the expected value are indicated by the dashed and solid yellow lines, respectively.**





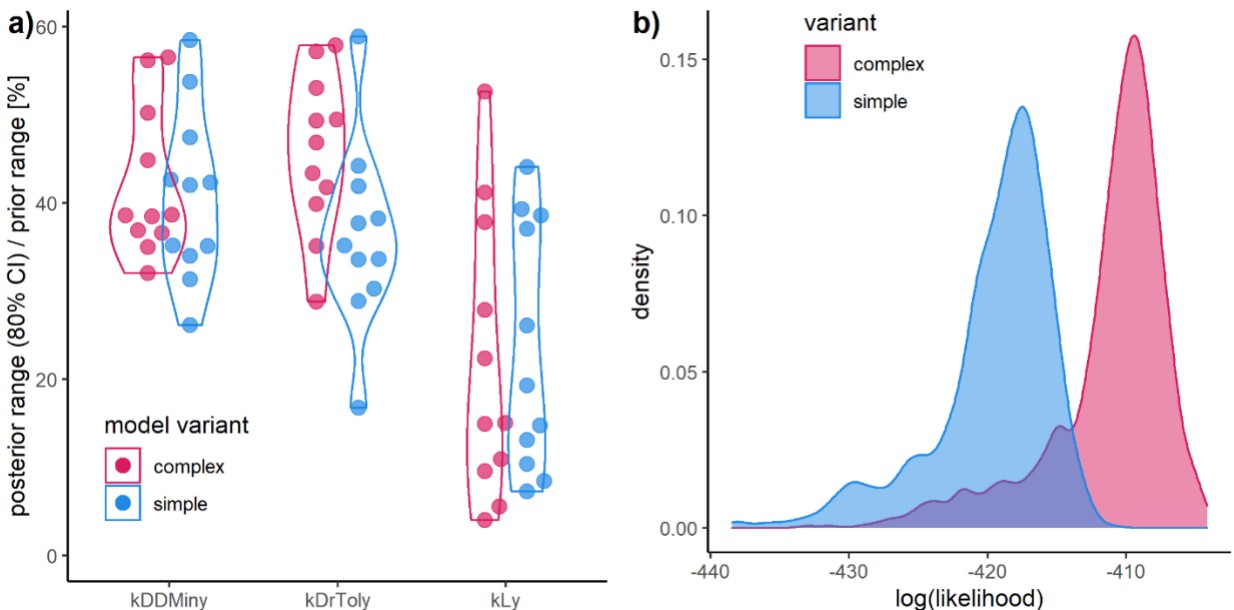

**Figure A2: a) Sensitivity of the parameters expressed by the percentage of the prior range that is covered by the 80% CI. b)**
**Likelihood distribution of the posterior simulations form the simple model (blue) and the complex model (red). The results for the**
**complex and the simple model are shown in red and blue, respectively.**




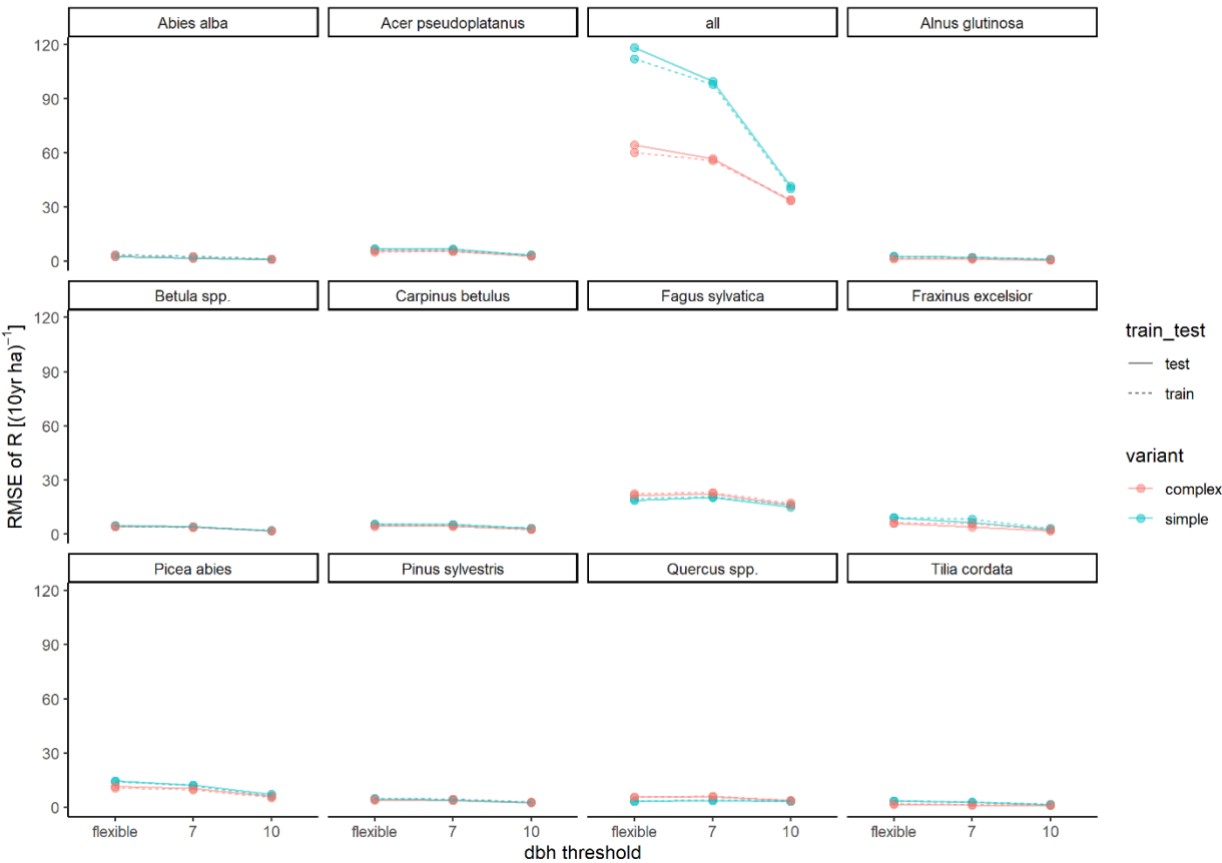

**Figure A3: Relationship between RMSE and DBH threshold. "flexible" refers to the individual DBH thresholds applied during model calibration. 7 and 10 cm comprise the subset of sites which had at least 7 or 10 cm DBH, respectively.**







**Figure A4: Root mean squared error (RMSE) of the difference between posterior predictions ($\widehat{R}$) and observations (R). Axis scaling differs between a) log10 scaled, b) varying x axis range with the trait-based approach TBA (default) RMSE, c) varying x axis only inverse calibration (ICA) posterior.**

**Table A1: Posterior values of the recruitment amount parameters kTrMax (complex model) and kEstDens (simple model), and the effects of DBH $\phi_{DBH}$ and plot area $\phi_A$ on dispersion. Where the outer values refer to the 80 % CIs and the middle values refers to the MPE. 10%CI | MPE | 90%CI**

|  | $\phi_{DBH}$ | $\phi_A$ | kTrMax | | kEstDens | |
|---|---|---|---|---|---|---|
|  |  |  | ICA | TBA | ICA | TBA |
| complex | -0.439\|-0.388\|-0.346 | -0.345\|-0.171\|-0.002 | 7372\|8762\|10210 | 50000 | - | - |
| simple | -0.473\|-0.405\|-0.345 | -0.261\|-0.120\|0.017 | - | - | 0.017\|0.022\|0.027 | 0.006 |

**Table A2: Species posterior parameter values for light requirements (kL$_y$), drought tolerance (kDrTol$_y$), and degree-days (kDDMin$_y$) including its default values. Where the outer values refer to the 80 % CIs and the middle values refers to the MPE. 10%CI | MPE | 90%CI. * Default values for Betula spp. are those of Betula pendula, default values for Quercus spp. are those of Quercus petraea.**

| species | kL$_y$ TBA | kL$_y$ ICA simple | kL$_y$ ICA complex | kDrTol$_y$ TBA | kDrTol$_y$ ICA simple | kDrTol$_y$ ICA complex | kDDMin$_y$ TBA | kDDMin$_y$ ICA simple | kDDMin$_y$ ICA complex |
|---|---|---|---|---|---|---|---|---|---|
| *Abies alba* | 0.03 | 0.02\|0.07\|0.12 | 0.01\|0.07\|0.15 | 0.23 | 0.10\|0.17\|0.25 | 0.11\|0.22\|0.34 | 641 | 810\|1058\|1301 | 844\|1072\|1292 |
| *Acer pseudoplatanus* | 0.05 | 0.11\|0.14\|0.16 | 0.05\|0.07\|0.10 | 0.25 | 0.15\|0.22\|0.30 | 0.08\|0.18\|0.29 | 898 | 927\|1148\|1365 | 305\| 677\|1091 |
| *Alnus glutinosa* | 0.2 | 0.18\|0.27\|0.37 | 0.13\|0.19\|0.25 | 0.08 | 0.19\|0.27\|0.35 | 0.02\|0.08\|0.14 | 898 | 661\| 997\|1325 | 576\| 827\|1092 |
| *Betula* spp.* | 0.5 | 0.25\|0.32\|0.38 | 0.07\|0.19\|0.33 | 0.16 | 0.11\|0.20\|0.29 | 0.06\|0.17\|0.28 | 610 | 512\| 892\|1265 | 427\| 752\|1054 |
| *Carpinus betulus* | 0.075 | 0.17\|0.20\|0.24 | 0.11\|0.14\|0.18 | 0.25 | 0.22\|0.29\|0.36 | 0.21\|0.30\|0.37 | 898 | 260\| 496\| 736 | 260\| 626\|1051 |
| *Fagus sylvatica* | 0.03 | 0.03\|0.05\|0.06 | 0.01\|0.02\|0.03 | 0.25 | 0.26\|0.32\|0.38 | 0.14\|0.24\|0.34 | 723 | 506\| 811\|1094 | 204\| 454\| 745 |
| *Fraxinus excelsior* | 0.075 | 0.12\|0.15\|0.19 | 0.05\|0.08\|0.12 | 0.16 | 0.26\|0.31\|0.37 | 0.08\|0.16\|0.26 | 980 | 443\| 737\|1040 | 293\| 635\| 996 |
| *Picea abies* | 0.05 | 0.10\|0.12\|0.14 | 0.03\|0.05\|0.08 | 0.15 | 0.04\|0.15\|0.28 | 0.16\|0.25\|0.33 | 385 | 416\| 603\| 782 | 208\| 461\| 720 |
| *Pinus sylvestris* | 0.4 | 0.20\|0.31\|0.42 | 0.11\|0.20\|0.30 | 0.37 | 0.02\|0.06\|0.09 | 0.15\|0.25\|0.34 | 610 | 828\|1147\|1419 | 180\| 432\| 719 |
| *Quercus* spp.* | 0.2 | 0.28\|0.39\|0.47 | 0.04\|0.05\|0.06 | 0.33 | 0.24\|0.31\|0.38 | 0.05\|0.11\|0.19 | 785 | 394\| 820\|1213 | 787\|1056\|1328 |
| *Tilia cordata* | 0.075 | 0.20\|0.30\|0.39 | 0.23\|0.33\|0.43 | 0.33 | 0.09\|0.15\|0.22 | 0.20\|0.29\|0.37 | 1339 | 260\| 512\| 752 | 246\| 487\| 735 |





**Table A3: Species dispersion parameter posterior values. Where the outer values refer to the 80 % CIs and the middle values refers to the MPE. 10%CI | MPE | 90%CI**

| species | complex | simple |
|---|---|---|
| Abies alba | 0.010\|0.018\|0.025 | 0.011\|0.019\|0.028 |
| Acer pseudoplatanus | 0.043\|0.072\|0.102 | 0.038\|0.064\|0.092 |
| Alnus glutinosa | 0.008\|0.018\|0.031 | 0.008\|0.018\|0.030 |
| Betula spp. | 0.001\|0.004\|0.008 | 0.001\|0.004\|0.009 |
| Carpinus betulus | 0.028\|0.052\|0.077 | 0.028\|0.048\|0.069 |
| Fagus sylvatica | 0.141\|0.199\|0.257 | 0.117\|0.161\|0.208 |
| Fraxinus excelsior | 0.035\|0.055\|0.080 | 0.029\|0.047\|0.064 |
| Picea abies | 0.087\|0.135\|0.182 | 0.081\|0.121\|0.165 |
| Pinus sylvestris | 0.013\|0.026\|0.040 | 0.009\|0.021\|0.032 |
| Quercus spp. | 0.000\|0.003\|0.006 | 0.001\|0.005\|0.010 |
| Tilia cordata | 0.026\|0.051\|0.080 | 0.019\|0.036\|0.054 |




**Table A4: Gelman-rubin diagnostics for the simple and complex model split by the two sets of in independent chains. For visual inspection of the chains see the traceplots in the dedicated file.**

| | simple model | | | | complex model | | | |
| --- | --- | --- | --- | --- | --- | --- | --- | --- |
| | chains 1-3 | | chains 4-6 | | chains 1-3 | | chains 4-6 | |
| | Point est. | Upper C.I. | Point est. | Upper C.I. | Point est. | Upper C.I. | Point est. | Upper C.I. |
| kEstDens/kTrMax | 1.001 | 1.002 | 1.034 | 1.050 | 1.004 | 1.015 | 1.073 | 1.090 |
| dispdbh | 1.000 | 1.000 | 1.232 | 1.657 | 1.002 | 1.003 | 1.274 | 1.761 |
| disppsize | 1.000 | 1.000 | 1.058 | 1.146 | 1.002 | 1.009 | 1.238 | 1.672 |
| kDDMiny_0 | 1.001 | 1.003 | 1.053 | 1.153 | 1.004 | 1.011 | 1.044 | 1.050 |
| kDDMiny_2 | 1.002 | 1.006 | 1.115 | 1.239 | 1.000 | 1.001 | 1.022 | 1.056 |
| kDDMiny_5 | 1.005 | 1.011 | 1.251 | 1.780 | 1.000 | 1.000 | 1.212 | 1.597 |
| kDDMiny_9 | 1.003 | 1.008 | 1.124 | 1.255 | 1.001 | 1.005 | 1.245 | 1.719 |
| kDDMiny_10 | 1.001 | 1.005 | 1.096 | 1.175 | 1.002 | 1.006 | 1.057 | 1.091 |
| kDDMiny_13 | 1.002 | 1.006 | 1.074 | 1.098 | 1.002 | 1.006 | 1.034 | 1.055 |
| kDDMiny_14 | 1.007 | 1.022 | 1.204 | 1.618 | 1.002 | 1.008 | 1.067 | 1.168 |
| kDDMiny_17 | 1.002 | 1.005 | 1.065 | 1.171 | 1.003 | 1.013 | 1.156 | 1.444 |
| kDDMiny_18 | 1.001 | 1.003 | 1.108 | 1.274 | 1.004 | 1.014 | 1.164 | 1.474 |
| kDDMiny_21 | 1.001 | 1.001 | 1.034 | 1.035 | 1.005 | 1.017 | 1.031 | 1.053 |
| kDDMiny_27 | 1.002 | 1.005 | 1.314 | 1.995 | 1.001 | 1.002 | 1.211 | 1.640 |
| kDrToly_0 | 1.004 | 1.010 | 1.080 | 1.088 | 1.003 | 1.007 | 1.025 | 1.030 |
| kDrToly_2 | 1.000 | 1.000 | 1.092 | 1.244 | 1.012 | 1.038 | 1.027 | 1.040 |
| kDrToly_5 | 1.004 | 1.011 | 1.138 | 1.325 | 1.000 | 1.001 | 1.101 | 1.228 |
| kDrToly_9 | 1.002 | 1.006 | 1.082 | 1.218 | 1.002 | 1.006 | 1.057 | 1.114 |
| kDrToly_10 | 1.001 | 1.004 | 1.108 | 1.273 | 1.003 | 1.009 | 1.324 | 2.116 |
| kDrToly_13 | 1.001 | 1.003 | 1.058 | 1.091 | 1.001 | 1.004 | 1.041 | 1.052 |
| kDrToly_14 | 1.001 | 1.002 | 1.222 | 1.681 | 1.000 | 1.001 | 1.071 | 1.167 |
| kDrToly_17 | 1.005 | 1.011 | 1.156 | 1.435 | 1.001 | 1.002 | 1.039 | 1.057 |
| kDrToly_18 | 1.004 | 1.007 | 1.284 | 1.981 | 1.002 | 1.008 | 1.176 | 1.504 |
| kDrToly_21 | 1.000 | 1.001 | 1.003 | 1.005 | 1.001 | 1.006 | 1.083 | 1.195 |
| kDrToly_27 | 1.001 | 1.006 | 1.182 | 1.530 | 1.002 | 1.006 | 1.228 | 1.676 |
| kLy_0 | 1.003 | 1.007 | 1.252 | 1.876 | 1.005 | 1.009 | 1.403 | 2.317 |
| kLy_2 | 1.002 | 1.005 | 1.138 | 1.280 | 1.002 | 1.003 | 1.030 | 1.039 |
| kLy_5 | 1.003 | 1.007 | 1.090 | 1.194 | 1.002 | 1.007 | 1.142 | 1.389 |
| kLy_9 | 1.007 | 1.014 | 1.190 | 1.559 | 1.000 | 1.001 | 1.037 | 1.060 |
| kLy_10 | 1.004 | 1.010 | 1.072 | 1.074 | 1.009 | 1.032 | 1.232 | 1.754 |
| kLy_13 | 1.002 | 1.004 | 1.110 | 1.235 | 1.004 | 1.012 | 1.112 | 1.289 |
| kLy_14 | 1.001 | 1.005 | 1.243 | 1.819 | 1.000 | 1.001 | 1.077 | 1.091 |
| kLy_17 | 1.000 | 1.001 | 1.034 | 1.093 | 1.001 | 1.002 | 1.037 | 1.062 |
| kLy_18 | 1.002 | 1.003 | 1.145 | 1.164 | 1.001 | 1.003 | 1.042 | 1.053 |
| kLy_21 | 1.006 | 1.013 | 1.210 | 1.643 | 1.001 | 1.004 | 1.066 | 1.115 |
| kLy_27 | 1.001 | 1.002 | 1.137 | 1.345 | 1.001 | 1.002 | 1.059 | 1.098 |
| disp_0 | 1.000 | 1.001 | 1.238 | 1.747 | 1.001 | 1.001 | 1.234 | 1.726 |
| disp_2 | 1.000 | 1.000 | 1.237 | 1.727 | 1.002 | 1.004 | 1.343 | 2.168 |
| disp_5 | 1.001 | 1.002 | 1.214 | 1.635 | 1.006 | 1.011 | 1.296 | 2.079 |
| disp_9 | 1.001 | 1.001 | 1.222 | 1.700 | 1.001 | 1.002 | 1.239 | 1.725 |
| disp_10 | 1.000 | 1.001 | 1.267 | 1.992 | 1.003 | 1.008 | 1.287 | 1.934 |
| disp_13 | 1.004 | 1.009 | 1.228 | 1.730 | 1.030 | 1.060 | 1.230 | 1.727 |
| disp_14 | 1.000 | 1.000 | 1.290 | 1.966 | 1.002 | 1.005 | 1.236 | 1.723 |
| disp_17 | 1.000 | 1.000 | 1.246 | 1.740 | 1.007 | 1.015 | 1.358 | 2.096 |
| disp_18 | 1.002 | 1.003 | 1.140 | 1.389 | 1.006 | 1.020 | 1.343 | 2.229 |
| disp_21 | 1.012 | 1.022 | 1.276 | 2.006 | 1.004 | 1.008 | 1.174 | 1.501 |
| disp_27 | 1.003 | 1.008 | 1.237 | 1.761 | 1.001 | 1.002 | 1.309 | 2.242 |



| disp_999 | 1.002 | 1.002 | 1.342 | 2.195 | 1.002 | 1.005 | 1.397 | 2.197 |




# Appendix B Model description

This Appendix provides a detailed description of the two model variants used in this study. It complements the description in the main script but only covers the regeneration module of ForClim. Note that this model incorporates some minor changes compared to the original formulations in Bugmann (1994) for the simple model and Huber et al. (2020) for the complex model. However, these changes do not affects the functioning of the model.

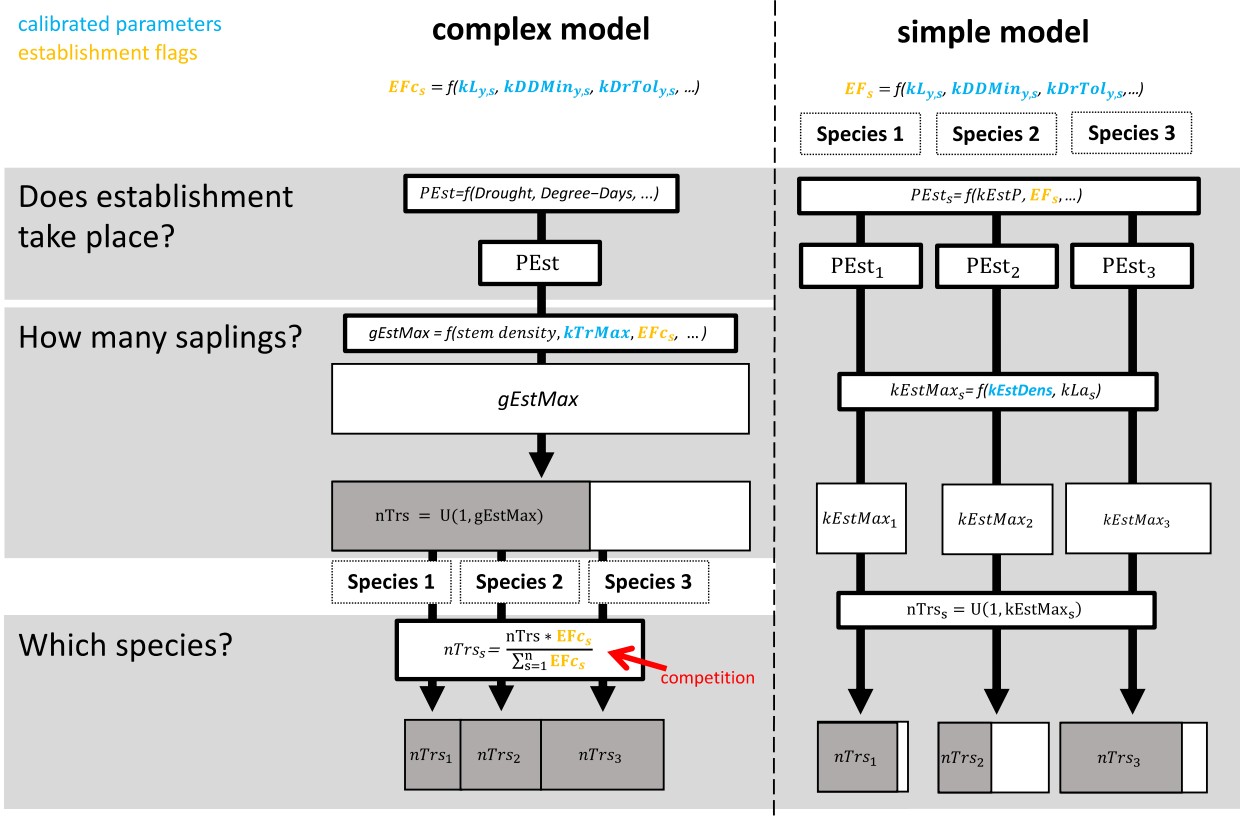

**Figure B1: Simplified visual representation of the simple and complex regeneration model variants of ForClim. As full description of the models is provided below.**



**Table B1: Description of ForClim model parameters that are considered for calibration.**

| Model variant | | Description | Default | Prior | Comment |
|---|---|---|---|---|---|
| simple & complex | $kDrTol_y$ | Species drought tolerance | 0.08-0.37 | 0.001-0.02 (0.001-0.4) | The species' drought tolerance and a drought index determine the EF for drought (Bugmann, 1994; Huber et al., 2020). |
| | $kDDMin_y$ | Species minimum degree-days | 385-1339 | 100-1500 | The species' minimum degree-day sum and degree-days determine the EF for temperature (Bugmann, 1994; Huber et al., 2020). |
| | $kL_y$ | Species light requirments | 0.03-0.4 | 0.001-0.5 | Species' light requirements and available light on the forest floor determine the EF for light (Bugmann, 1994). |
| complex | kTrMax | Maximum Number of Trees per ha | 30000 | 500 – 50000 | Maximum number of trees per ha refers to the number of trees regardless of the species. It also includes the trees that are already present on the patch (Huber et al., 2020). |
| simple | kEstDens | Maximum Establishment Density [trees/(m²·yr)] | 0.006 | 0.001- 0.2 | Maximum tree establishment density is defined per species (Bugmann, 1994) |



**Table B2: Description of ForClim all model variables and parameters used in this study. Calibrated parameters are explained in more detail in Table B1.**

| group | variable | unit | description | model | reference |
|---|---|---|---|---|---|
| establishment flags | ALEF | boolean | Light availability establishment flag | simple | Bugmann, 1994 |
| | ALEFc | %/100 | continuous light availability establishment flag | complex | Bugmann, 1994 |
| | BPEF | boolean | Browsing pressure establishment flag | simple | Bugmann, 1994 |
| | BPEFc | %/100 | continuous browsing pressure establishment flag | complex | Bugmann, 1994 |
| | DDEF | boolean | Degree-days establishment flag | simple | Bugmann, 1994 |
| | DDEFc | %/100 | continuous degree-days establishment flag | complex | Bugmann, 1994 |
| | SMEF | boolean | Soil moisture establishment flag | simple | Didion et al., 2009a |
| | SMEFc | %/100 | continuous soil moisture establishment flag | complex | Didion et al., 2009a |
| | WTEF | boolean | Winter temperature establishment flag | simple | Bugmann, 1994 |
| | WTEFc | %/100 | continuous winter temperature establishment flag | complex | Bugmann, 1994 |
| general parameter | kDDLL | °C·d | annual DD below which DDsum prevents establishment | complex | Bugmann, 1994 |
| | kDDUL | °C·d | annual DD above which DDsum is not reducing kTrMax | complex | Bugmann, 1994 |
| | kDrLL | %/100 | mDrAn bewlow which drought is not reducing kTrMax | complex | Bugmann, 1994 |
| | kDrUL | %/100 | mDrAn above which drought prevents establishment | complex | Bugmann, 1994 |
| | kEstDens | m-2·yr-1 | Maximum number of tree establishment density | both | Bugmann, 1994 |
| | kEstP | %/100 | general probability of regeneration | both | Bugmann, 1994 |
| | kPatchSize | m2 | size of a forest patch | both | Bugmann, 1994 |
| | kTrMax | ha-1 | Maximum number of trees per ha | both | Huber et al., 2020 |
| | kDDMin | °C·d | minimum degree-day sum for adults | both | Bugmann, 1994 |
| | kDDMiny | °C·d | minimum degree-day sum for regeneration | both | - |
| | kDrTol | %/100 | drought tolerance for adults | both | Bugmann, 1994 |
| | $kDrTol_y$ | %/100 | drought tolerance for regeneration | both | - |
| | $kL_a$ | [1-9] | adult light requirements as proxy for seed production | simple | Risch et al., 2005 |
| | $kL_y$ | %/100 | light requirements of regeneration | both | Bugmann, 1994 |
| state variable (regeneration) | EFc | %/100 | minimum continuous establishement flag | complex | Huber et al., 2020 |
| | EFSum | - | sum over all continuous establishment flags | complex | - |
| | gEstMax | - | annual potential maximum number of established trees | both | Bugmann, 1994 |
| | kEstMax | - | maximum regeneration for a species | simple | Bugmann, 1994 |
| | nTrs | - | the number of trees that are being recruited | both | Bugmann, 1994 |
| | PEst | %/100 | realized regeneration probability | both | - |
| | uEFMax | %/100 | helper variable to ensure that Efc is not smaller than 0 | complex | - |
| | ukEstP | %/100 | helper variable to calculate PEst in the complex model | complex | - |
| state variable (site) | gDD | °C·d | mean annual degree days from weather generator | both | Bugmann, 1994 |
| | gDr | %/100 | mean annual drought index from weather generator | both | Bugmann, 1994 |
| | gRedFac | %/100 | overall reduction factor | complex | Huber et al., 2020 |
| | gRedFacDD | %/100 | reduction of kTrMax caused by degree days | complex | Bugmann, 1994 |
| | gRedFacDI | %/100 | reduction of kTrMax caused by drought index | complex | Bugmann 1996 |
| | mDDAn | °C·d | mean annual degree days from weather generator | complex | Bugmann 1996 |
| | mDrAn | %/100 | mean annual drought index from weather generator | complex | Bugmann 1996 |
| state variable (stand) | gAL0 | %/100 | avalaible light at the forest floor | both | Bugmann 1996 |
| | Trs | - | N of trees of all species in current simulation year | both | Bugmann 1996 |





**Step 1 – Does regeneration take place?**

First, it is determined whether regeneration takes place at all in any given year; this is done for each species on each patch. The probability of regeneration (kEstP) was set to 2% for the calibration, lower than the default value of kEstP of the default model (trait-based approach) (10%). This was done to reduce the number of cohorts and thus computational costs. The reduction, however, does not reduce effective regeneration because it can be compensated by the parameters for the

regeneration amount (kEstDens and kTrMax; cf.





Table B2 and eq. eq. A3 and eq. A4), in the sense that a reduction of kEstP by a factor of 1/5, for example, can be compensated by an increase of kEstDens or kTrMax by a factor of 5.

$$PEst_s = kEstP * WTEF_s * ALEF_s * BPEF_s * DDEF_s * SMEF_s$$

$$PEst_s = \begin{cases} 1, & U(0,1) < PEst_s \\ 0, & else \end{cases}, \qquad \text{eq. A1)}$$

where U is a function that draws a random number from a uniform distribution ranging from 0 to 1. The complex model determines whether regeneration takes place (PEst = 1) or not (PEst = 0) regardless of the species but depending on the site

factors degree-days (mDDAn), and drought (mDrAn). Specifically, PEst is calculated with

$$kDrLL = 0.1; \ kDrUL = 0.5$$

$$kDDUL = 1225; \ kDDLL = 0$$

$$gRedFacDI = \frac{mDrAn - kDrLL}{kDrUL - kDrLL}$$

$$gRedFacDD = 1.0 - \frac{mDDAn - kDDLL}{kDDUL - kDDLL} \qquad \text{eq. A2)}$$

$$gRedFac = Max(gRedFacDI, gRedFacDD)$$

$$ukEstP = kEstP * (1 - gRedFac)$$

$$PEst = \begin{cases} 1, & U(0,1) < ukEstP \\ 0, & else \end{cases}.$$

**Step 2 – How much regeneration?**

Second, the number of new trees is calculated.

The **simple model** simulates the maximum regeneration amount per species based on the establishment intensity parameter

(kEstDens), but also based on $kLa_s$ which is used as a proxy for seed production (Huber et al., 2020; Risch et al., 2005). Specifically, the maximum regeneration $kEstMax_s$ for species s is calculated with

$$kEstMax_s = PEst_s * kEstDens * kPatchSize * kL_{a,s} , \qquad \text{eq. A3)}$$

where kPatchSize is patch size, which was set to 800 m².

In the **complex model,** regeneration amount is regulated by the continuous establishment flag (EFc) of the most suitable species, site factors (see calculation of gRedFac in eq. A2), and the regeneration intensity parameter kTrMax, Specifically

the regeneration amount over all species (nTrs) is calculated with



$$EFc_s = MIN(WTEFc_s, ALEFc_s, BPEFc_s, DDEFc_s, SMEFc_s)$$

$$uEFMax = MAX(0, EFc_s)$$

$$gEstMax = PEst * uEFMax * \left( \frac{kTrMax * (1 - gRedFac)}{10000 * kPatchSize} - Trs \right) \qquad \text{eq. A4)}$$

$$nTrs = U(1, gEstMax)$$

**Step 3 – What species?**

Third, the final number of new trees for each species ($nTrs_s$) is determined.

In the **simple model**, for each species s a random number between 1 and $kEstMax_s$ is drawn with

$$nTrs_s = U(1, kEstMax_s). \qquad \text{eq. A5)}$$

In the **complex model**, $nTrs_s$ is calculated for each species s based on its suitability for regeneration ($EFc_s$) and relative to the suitability of all other species (EFSum) with

$$EFSum = \sum_{s=1}^{n} EFc_s$$

$$nTrs_s = \frac{nTrs * EFc_s}{EFSum} \qquad \text{eq. A6)}$$

Ultimately, $nTrs_s$ denotes the number of trees with a DBH of 1.27 cm in a new cohort per species s on one patch and serves as a basis for calculating the likelihood (see below). After a new cohort has established, it follows the rules of adult growth and mortality, which remain unchanged throughout this study by keeping all model parameters that directly affect trees above a DBH of 1.27 cm at their default (Bugmann, 1994; Huber et al., 2020).

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
