# Peer review of "Inferring the tree regeneration niche from inventory data using a dynamic forest model"

_EGUsphere, 2023_

## Author Comment (AC1)

Dear Editors and Reviewers,

We are grateful for the constructive and positive feedback, which has significantly contributed to the improvement of our manuscript.

Both reviewers raised concerns about the readability and clarity of the manuscript, and we agree. In particular, the introduction needed improvement to make the text more accessible to a wider audience. At the same time, we acknowledge that there was a lack of coherence in various parts of the text. Thank you for your valuable comments and suggestions to resolve these issues. We thoroughly reviewed each comment and made appropriate revisions to the manuscript. The text is now easier to understand and the introduction is more coherent.

The main improvements made are
- Extensive improvements to the introduction by removing much of the text on concepts and ideas not directly related to our study. We have also placed more emphasis on explaining the context and motivation behind our methodological approach.
- We have moved Figure B1 and Table B1 to the methods section as we believe they provide important information for understanding our approach.
- We resolved an issue with the definition and use of the terminology regarding the likelihood.

We believe that these changes have significantly improved the accessibility and clarity of the manuscript.

A point-by-point response to the reviewers' comments, is attached. The revised manuscript will be uploaded for your consideration. We appreciate the opportunity to refine our work and look forward to any further comments you may have.

Thank you for your attention to our submission and for the valuable insights you have provided.

Best regards,
Yannek Käber on behalf of all co-authors

**Point-by-point response**

**Reviewer 1**

**General comments**

**General comment paragraph 1**
Within the framework of the ForClim forest model, the authors present a computationally intensive study that derives, for two regeneration models ("simple" and "complex"), based on recruitment rates observed in extensive forest inventory data, Bayesian estimates and credible intervals of parameters representing the regeneration intensity (kEstDens or kTrMax, depending on the model) and species-specific regeneration (or "young tree") parameters representing shade, temperature and drought tolerance (kLy, kDDMiny, kDrToly, respectively). They then compare these estimates – and their effect on the performance of subsequent simulations – (1) to earlier estimates based on ecological knowledge and (2) between the two models, and they provide a detailed discussion of the results.

**General comment paragraph 2**
The general approach is sound and well-thought-out, and the manuscript appears to have been carefully prepared, with useful and readable appendices. The full code and data are

provided, although, due to the complexity of the analysis (several thousand files) I cannot, in the scope of this referee comment, assess the quality or correctness of the implementation. Also, the study is relevant because the justification of parameter values chosen in simulation models is, aside the model structure, often one of their weakest points, and a careful analysis such as the one presented here can help shed light on these questions.

**Response**: Thank you for seeing the value in our study. We understand that the code and data is extensive. However, we hope that the description of the key technical aspects still allows a good understanding of the implementation of our approach.

**General comment paragraph 3+4**

A difficulty I see for many potential readers is the high level of technicality in the presentation, which, together with some smaller issues (like wording that is not always very concise or clear, and long lines) makes the manuscript a bit hard to read. This is probably in part due to the complexity of the model, but given this complexity I believe there is still room for improvement.

To make the manuscript accessible to a wider audience, I therefore suggest that the authors try to read the manuscript the from the point of view of an uninitiated reader and revise it so as to present important arguments in the main text in a concise and not too technical language. Some diagrams or tables might be of help here. In fact, the appendices already contain much of that material. For example, it would probably be a good idea to present the table with the relevant parameters (see Table B1) early on in the main text to help readers see through the various k... variables, which otherwise, on a first reading, are just overwhelming.

**Response**: We agree. In order to improve the readability and to make the presentation of our study more accessible, we have improved the manuscript in several aspects. Specifically, we took special care of the following issues:

- We improved the abstract by making minor corrections for clarification.
- We made extensive improvements to the introduction.
- We moved Figure B1 to the methods section of the main text.
- We moved Table B1 to the methods section of the main text.

There are also a number of smaller comments and improvements that are listed below.

**Specific comments**

**ABSTRACT**
**Line 12**: "determine": Replace with "determines".
**Response**: Done.

**Line 23**: "in": Remove.
**Response**: done

**INTRODUCTION**

**Lines 44-45**: This is confusing as it gives the impression that the regeneration niche affects the entire lifespan of a tree. My guess is that "one adult" refers to a dying tree and the "new adult" refers to a tree that just reached maturity, but this distinction is not clear from the way it is written. It also suggests there are well-defined generations.
**Response**: We have replaced the definition with a direct quote from Grubb to clarify its origin and meaning. This should also resolve the issue of defining "one adult" and "new adult"

**Line 49**: "niches of young trees and adults": This distinction is important, but isn't the niche of a young tree already an oversimplification since, clearly, the niche very much depends on the life stage of the young tree?
**Response**: We mentioned this aspect to place the paper in a broader ecological context. However, also based on R2's desire to focus more on methodological aspects, we removed the more detailed paragraph and now mention the concept of ontogenetic shifts in environmental preferences only very briefly to acknowledge the limitations of the concept (newL44). This also helps to make the introduction more concise.

**Line 64**: I would assume that mortality is one of the key determinants of regeneration because most trees do not survive a very young age.
**Response**: We agree. In fact, many DFMs appear to compensate for very high regeneration rates with excessive mortality in early tree growth. This is discussed further in L485 (newLXXX TODO). We have added "... of adult trees" to make this difference transparent. newL49

**Line 78**: How does this definition of the regeneration niche fit to the abstract definition in lines 44-45? Which relevant aspects are possibly missing in this more specific definition? Also, I do not understand "among others", which also appears to be in the wrong place.
**Response**: Thank you for pointing this out. We agree that it is confusing. In the context of the ForClim model we should not redefine the niche. Instead we have reformulated it by using the term "is derived from" instead of "defined". newL82
"among others" refers to the other variables and factors not explicitly mentioned here. We have changed this to "among other factors". We hope this makes it clearer. newL81-82

**Line 81**: "Indeed ... DFMs:": This sentence is not clear; maybe it is not needed.
**Response**: To make it clearer and easier to understand, we have removed this sentence and reworded the following sentences. newL86-87

**METHODS**

**Lines 110-111**: What is the reason for the implausible measurements? How can we be sure the 696 plots not removed (of a total of 865) have correct measurements?
**Response**: The plots were removed based on certain plausibility thresholds for annual changes in dbh growth. For example: we specified the range of -0.1 to 2 cm as a "plausible" range of annual dbh change.

We acknowledge that we cannot be sure 100 % that the measurements of the remaining plots are correct, nor can we be 100% sure that a tree hasn't grown by more than 2cm or

has even shrunken. However, we believe that overall, our approach will mostly filter wrong values that thus reduces distortions that originate from gross observation errors that are ecologically impossible.

We added some additional information and hope that this makes it clear. newL120-122

**Line 192**: Remove "(".
**Response**: done

**Line 225**: P(...) should probably be removed here.
**Response**: done

**Lines 235-237**: There appears to be some confusion here. Assuming the recruitment rates for all species and plots are independent random variables (which might need some justification), the likelihood, being the probability density function evaluated at the observed data, is multiplicative. Taking logarithms to transform multiplication into addition we arrive at the log-likelihood, which, consequently, is additive. So taking sums of log-likelihoods results in a joint log-likelihood, not likelihood. In fact, I would suggest to explicitly include "log" in all expressions that are supposed to represent log-likelihoods (i.e. write log P(...)).

**Response**: Thank you - we completely agree with all that is said here. Whether one calls the final likelihood a joint likelihood or a likelihood seems to us a matter of the viewpoint. It is common practice to refer to the joint likelihood of all data points in a regression as the likelihood of the data. We agree that it should be made clear whether we talk about L or LogL. We now added a log before the likelihood where summation takes place.

**Lines 235-237 (continuation)**: Also, while the representation of the densities determines a multiplicative constant in the likelihood and hence an additive constant in the log-likelihood, a rescaled log-likelihood is no longer a log-likelihood, although it can, of course, be used in the way the authors indicate. (Rescaling the log-likelihood corresponds to changing the base of the logarithm, but in statistical theory log usually means the natural logarithm.) In line 235 the authors need to decide whether this is about the intermediate sums over the plots for each species or about the total sum.

**Response**: We agree that an ad-hoc rescaling such as applied here is not justifiable in the standard statistical likelihood framework. We believe, however, that there are good reasons for a "conservative" re-scaling such as the one we do here in the case of stochastic simulation models, and this was also done in other papers.

The effect of re-scaling the likelihood with a factor <1 is that it increases the uncertainty in the posterior estimates and makes the MCMC less prone to getting stuck due to model stochasticity, while it usually does not change the MAP (i.e. the "best fit"). This is what we man by "conservative"

Apart from the better convergence, the wider posterior implicitly accounts for model error, see "Oberpriller, J., Cameron, D. R., Dietze, M. C., & Hartig, F. (2021). Towards robust statistical inference for complex computer models. Ecology Letters, 24(6), 1251-1261" and

also non-independence in the data (which you mentioned) that would be difficult to treat in a formal way.

We acknowledge that there is no clear justification for why the factor should be 1/12, but based on the arguments above, we believe it is in the right order of magnitude if we translate this as saying: based on model error and non-independence, we believe that we have just 1/12 of the number of data points that we formally seem to have.

We have now included these arguments also more clearly in the manuscript. newL275-284

**Line 325**: "following": This seems to be the wrong word.
**Response**: "following" was removed.

**Line 362**: Presumably the p-values are not meaningful here because the implied test of independence of the TBA and ICA features rests on the assumption that the (TBA, ICA) pairs for different species are exchangeable (for example, resulting from independent repetitions of the same experiment), which is clearly not satisfied here.
**Response**: We agree. For this reason we removed the p-values from the graph to avoid misinterpretation. newL384 (new Figure 3)

**Line 491**: "estimates": Does this mean the credible intervals?
**Response**: You are correct. We mean the credible intervals.

**Line 584**: "likelihood": This needs some clarification because likelihoods exist in any statistical model. Possibly the authors here refer to an exact computation in a part of their analysis as opposed to some approximate procedure.
**Response**: Thank you for pointing this out. This was not clear from the way we wrote it. We now refer to "our approach to derive the likelihood" and also to the corresponding equations in the methods section.

**Line 608**: Remove ".".
**Response**: done

**Line 910**: "affects": Remove "s".
**Response**: done

**Line 926**: eq. A1: Having two variables PEst s and PEsts whose names differ by a space in the index only is probably not a good idea. (Also, "×" might be nicer than "*".)
**Response**: PEst_s is in fact one variable. In the second part it is only updated based on the if else statement. We also changed * to x throughout the manuscript.

**Reviewer 2**

**General Comments**

**General comment paragraph 1**

The aim of this study was to validate and compare two regeneration modules (simple and complex) of the ForClim model using an inverse calibration and trait-based approach. They found that parameters determining species' light requirements are consistent between trait-based and calibrated values for both model variants. Parameter values were consistent between the two models but changed during the calibration process, with temperature and drought exhibiting the highest uncertainties.

**General comment paragraph 2**

The methodology and results appear robust, and I do not see any methodological issues. The methodology is innovative, and model validation is crucial. However, more effort is needed to better understand the work. The methodology is quite complex for readers in ecology, and the introduction does not adequately introduce the concepts necessary for understanding the methodology. A simplified conceptual diagram of the methodology could greatly enhance the main text. For instance, including Figure B1 in the main text, along with both calibration methods, would be a valuable addition. The results and discussion were well-written and understandable.

**Response**: We agree. The methodology is complex but should be understandable to a wide audience. We have therefore moved Figure B1 and Table B1 to the main text. Thank you for this suggestion, which improved the presentation and made the manuscript more accessible.

**General comment paragraph 3**

Introduction: More work is needed for the introduction; it does not allow us to understand correctly and lacks the keys to comprehend the objectives. For instance, more details should be provided to understand inverse calibration. The initial paragraphs focusing on niches do not contribute to understanding; on the contrary, they distract the reader from more crucial matters. As your article leans more towards methodology, the introduction should present the philosophy behind the methodology. This is especially important in the context of a changing climate. Additionally, elucidate more on how DFMs model regeneration. The sentences are often too complex for effective reading.

**Response**: We agree. The focus of the introduction was not clear enough. We admit that it was somewhat unclear what the key concepts and methods used in our study were. The structure of the introduction is now clearer. Specifically, we took special care of the following issues:
- We improved the abstract by making minor corrections for clarification.
- We made extensive improvements to the introduction.
- We moved Figure B1 to the methods section of the main text.
- We moved Table B1 to the methods section of the main text.

**Minor comments**

**INTRODUCTION**
**Line 37**: Why is it particularly important?
**Response**: We now mention examples:
"In this context, tree regeneration is particularly important because of its key role in species range shifts (McDowell et al., 2020), and forest resilience to climate change (i.e., reorganisation after disturbances; Seidl and Turner, 2022)." newL34-36

**Line 44**: The regeneration niche is defined as the set of factors for successful replacement. It's odd to have an increase when discussing the niche.
**Response**: We replaced the definition by a direct quote of Grubb to make its origin and meaning clearer. This change makes formulation easier to understand. newL45-47

**Line 49**: I wouldn't use affirmations such as: "Differences between the niches of young trees and adults are evident."
**Line 51**: Ontogenetic niche shifts are differences between the niches of young trees and adults. One cannot explain the other if the entities are the same.
**Line 53**: "Together" refers to what? The same goes for line 45.
**Response**: We mention this aspect to frame the paper in a broader ecological context. We removed the more detailed paragraph and now mention the concept of ontogenetic shifts in environmental preferences only very briefly to acknowledge the limitations of the concept (newL44). This also helps to make the introduction more concise, puts a stronger focus on the methodological aspects, and avoids distracting the reader (as you suggest in one of your major comments).

**Line 57**: What are the issues?
**Response**: We reformulated this sentence to "… assess how simulated natural regeneration is related to real-world observations" to clarify what we are actually doing. newL65

**Lines 58-59**: I didn't understand the second part of the sentence: "which puts the quantified effects in the context of specific processes."
**Response**: We added an example in the following sentence to explain what we mean. "For example, species' shade tolerance estimates will only be constrained by the actual available light and not by any other confounding factors." newL66-67

**Line 58**: Replace "represent" with "model"?
**Response**: done

**Line 64**: "Regeneration in most DFMs is captured relatively simply." Be more specific.
**Response**: We now specify "simple" and "complex" and also add specific examples. We also explain the general function of regeneration models in DFMs, also to make the methodological approach more accessible. newL53-61

**Lines 73-75**: "Specifically, trade-offs between meaningful observations for key small-scale processes and the coverage of wide abiotic gradients impede a comprehensive analysis

across the stages of tree regeneration (Clark et al., 1999)." I didn't understand; give examples instead.

**Response**: We now provide some examples for "small-scale processes" and replaced "abiotic gradients" with "macroclimatic gradients or scales at which dispersal takes place". newL76-80

**Lines 79-84**: I did not understand the end of this long sentence: "which is explicitly modeled are expected to have more predictive power than traits that act on multiple processes which are not considered individually." Keep your sentences short and comprehensible.

**Response**: We restructured this sentence by splitting it up. In addition, we provide some examples. newL83-89

**Line 86**: I didn't understand this sentence: "In ForClim, the interplay between the traits of multiple species is implemented in a simple variant of the regeneration model that does not consider competition and a more complex model variant that includes it."

**Response**: We now provide more information regarding both model variants. newL91-95

**Line 90**: The regeneration process cannot be synthesized by only seedling and sapling growth rates; the mortality and density of seedlings are also important components of regeneration.

**In line 90**, "it remains unclear whether the simulated regeneration processes (i.e., ingrowth rates) are actually in agreement with observational data," but in line 92, "we evaluate possible reasons for mismatches between process formulations and observations." You acknowledge the uncertainty regarding whether the simulated processes align with observations, and then you evaluate the reasons for the mismatches. However, these two paragraphs do not establish a clear connection.

**Response**: Thank you for pointing this out. The connection between these paragraphs should now be much clearer.
We elaborate a bit more on the actual research gap this paper is addressing. newL96-101

**METHODS**
**Lines 147-148**: I did not understand this sentence. And it's unfortunate because it is central to the understanding of the article.
**Response**: We improved the sentence, added an example and also refer to the more detailed explanation of the idea behind the establishment flags further below in the text. newL160-166

**RESULTS**
**Lines 321-322**: This kind of sentence is unnecessary.
**Response**: The sentence was removed.

**Line 354**: The explanation for the grey rectangle regarding priors is unclear.
**Response**: We improved the explanation and hope that it is more clear now. newL382-383

**Acceptance Email for Díaz-Yáñez et al.**

Note that the cited paper from Díaz-Yáñez et al. is now accepted in Ecosphere and will be published within the next few weeks.

Díaz-Yáñez, O., Käber, Y., Anders, T., Braziunas, K. H., Bruna, J., Fischer, S., Hetzer, J., Hickler, T., Hochauer, C., Lexer, M., Lischke, H., Mahnken, M., Mairota, P., Merganicova, K., Mette, T., Morin, X., Rammer, W., Scheiter, S., Scherrer, D., & Bugmann, H. (in press). *Tree regeneration in models of forest dynamics: A key priority for further research*.

---

## Author Response (AR2)

Dear Editors and Reviewers,

We are happy that you provisionally accepted our manuscript for publication.

All final issues raised were resolved in the new manuscript version. The specific changes are documented below. We also updated the doi for the Zenodo repository since minor changes were made to the figure in the last revision. This required an update of the scripts in the repository. The updated Zenodo repository will be published as soon as the paper is published. This allows me to also add a proper reference to the published paper in the repository.

Note that the cited paper from Díaz-Yáñez et al. is still in press and we are waiting for the DOI, which we will probably get within a few days. Therefore we shall be able to include the DOI during typesetting.

Please let us know if we can assist the publication process in any way.

Thank you for the very constructive and smooth review processes.

Best regards,
Yannek Käber on behalf of all co-authors

**Point-by-point response**

**Reviewer 1**

In the revised version, the authors considerably improved the manuscript with regard to the criticisms of my earlier review. In particular, the introduction has been significantly improved, thus enabling readers to get a foothold in the article. In addition, some inconsistencies have been removed.

Overall, I still consider this a highly technical manuscript, which, however, is for the most part well worked out and definitely of interest to the modelling community; it should therefore be published.

I list some smaller technical issues:

**Line 274**: This sentence, as written, still makes no sense logically because $\log[Ps(ys \mid \theta)]$ is not a sum over s. I suspect the authors want to say something like this: "Summing the expressions in 10) over all plots i, we obtain the log-likelihood $\log[Ps(ys \mid \theta)]$ for a species s, and further summing this over s we arrive at the joint log-likelihood $\log[P(y \mid \theta)] = \sum \log[Ps(ys \mid \theta)]$, which we here analyze in a scaled form
$\log[P(y \mid \theta)]/Nspecies = \sum \log[Ps(ys \mid \theta)]/Nspecies$ ..."
As I understand it, the scaling is merely a technical point in algorithm (the authors added some explanation here) and therefore not an issue. As indicated, I recommend to reserve probability expressions such as $\log[P(y \mid \theta)]$ for the unscaled version.

**Response**: We changed the explanation of the likelihood to resolve these issues. For this we included the suggested formulation and made small changes to the equations to avoid using the probability expression.

**Line 389**: "therefor": Should probably be "therefore".
**Response**: fixed.

**Line 936** (equation A1): While an assignment expression like x = x + 1 is a common idiom in many programming languages, such "equations" are unacceptable for a mathematical presentation. The authors should therefore use different variable names for the preliminary PEsts and the updated PEsts (or perhaps an additional index) to make the distinction clear.
**Response**: We changed the name of the first variable to PEstEF_s. This makes the equation also valid for a mathematical presentation.